# PROOF-AUGMENTED RETRIEVAL AND REASONING: SUPERVISING LANGUAGE MODEL FOR KNOWLEDGE GRAPH COMPLETION WITH INTERPRETABLE LINK PREDICTORS

## ABSTRACT

We propose Proof-Augmented Retrieval and Reasoning (PARR), a general framework for training language models for Knowledge Graph Completion (KGC). PARR leverages proof paths sampled from interpretable link predictors (1) to augment retrieval database for enhanced sub-graph retrieval, (2) as ground-truth signals to train a REWRITER LLM for KG-based query rewritings, and (3) as a mean to "distill" the structural knowledge captured from pre-trained link predictors with structural prior to a REASONER LLM through chain-of-thoughts. PARR achieves state-of-the-art performance across multiple KGC datasets under both transductive and inductive settings, while being generative, scalable, and interpretable.

## 1 INTRODUCTION

Knowledge Graph Completion (KGC) is a fundamental task in machine learning that seeks to infer missing relations between entities in large-scale knowledge graphs. KGC enables downstream applications in diverse domains such as recommendation systems, scientific discovery, and healthcare.

Most traditional KGC methods fall into two categories. Knowledge graph embedding (KGE) models (Bordes et al., 2013; Trouillon et al., 2016; Sun et al., 2019) learn vector representations of entities and relations for efficient link prediction. GNN-based approaches (Vashishth et al., 2020; Zhu et al., 2021; 2023) leverage message passing to capture local graph structure. These models are lightweight and effective, but real-world knowledge-intensive domains often prioritize accuracy, interpretability, and interactive reasoning over raw inference latency. Moreover, both KGE and GNN models are inherently discriminative: predictions are made by scoring candidate entities, offering limited explainability and weaker generalization outside training distributions.

Large Language Models (LLMs) (lla, 2024; Yang et al., 2024) offer appealing generative and conversational abilities, but they do not naturally excel at KGC. Unlike KGE or GNN models built around structural priors, LLMs operate in the much larger space of natural language. Despite their internal knowledge, augmenting KGs with LLM-generated background information yields little improvement (Jiang et al., 2024). By contrast, methods such as NBFNet and ULTRA (Zhu et al., 2021; Galkin et al., 2024) achieve strong link prediction solely by modeling relational and structural patterns, without explicit entity embeddings. This suggests that background knowledge of entities is not the key ingredient for KGC. Prior attempts to directly fine-tune LLMs on KGC also underperform traditional link predictors significantly. (Yao et al., 2025; Zhu et al., 2024). Another line of work repurposes LLMs as discriminative encoders, trained with standard KGE objectives such as binary cross-entropy loss and negative sampling (Yao et al., 2019; Guo et al., 2024). While these methods improve accuracy, they reduce LLMs to specialized link predictors, sacrificing the generative and interactive capabilities that make them attractive for explainable KGC.

In this work, we identify three key challenges that have limited prior LLM-based approaches:

*(1) Retrieval.* Unlike natural language retrieval, where information is naturally grouped into paragraphs or documents, knowledge graphs lack inherent clustering patterns. Moreover, as there is no ground-truth retrieval labels for KGs, prior LLM-based link predictors often rely on simple

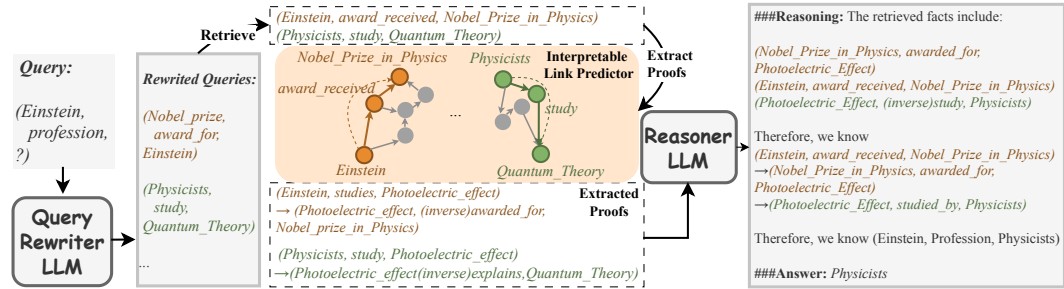

Figure 1: Overall pipeline of PARR. Given a query such as *(Einstein, profession, ?)*, the Rewriter LLM produces a set of semantically varied yet logically related sub-queries (e.g., *(Nobel_prize, award_for, Einstein)*) and *(Physicists, study, Quantum_Theory)* to broaden retrieval. Pre-trained interpretable link predictors (e.g., NBFNet (Zhu et al., 2021)) then provide proofs for retrieved facts, enabling targeted subgraph retrieval. The collected evidence is passed to the REASONER LLM to derive the final answer.

similarity-based strategies. For example, KICGPT (Wei et al., 2023) retrieves all triplets sharing the same head or tail entity, which is constrained to local neighbors while missing global context.

*(2) Reasoning.* Although LLMs demonstrate strong general reasoning ability, they are prone to hallucination in KG reasoning when exposed to noisy or irrelevant retrievals. As shown in Figure 3, LLM performance drops sharply as irrelevant context accumulates, highlighting the need for more robust reasoning mechanisms that can operate under imperfect retrieval.

*(3) Structural understanding.* Existing LLM-based KGC methods (Yao et al., 2025; Zhu et al., 2024) often reduce link prediction to a one-step task: given a subject and relation, directly predict the object. However, this formulation makes it difficult for LLMs to capture the underlying relational and structural patterns of the graph.

To address these challenges, we introduce Proof-Augmented Retrieval and Reasoning (PARR). PARR leverages proof supervision from interpretable link predictors to fine-tune native LLM-based generative link predictors, enhancing both retrieval and reasoning in three complementary ways:

*(1) Proof-augmented retrievals.* At retrieval time, we retrieve not only relevant triplets but also their associated proofs. Proofs serve as natural clusters that capture both local and global subgraph structure, yielding more comprehensive yet compact retrievals.

*(2) Proof-guided Query Rewriting.* Proofs could also act as pseudo ground-truth signals for retrieval. Inspired by query rewriting in natural language RAG (Ma et al., 2023), we fine-tune a REWRITER LLM to decompose queries into sub-queries, improving non-local retrieval and recall.

*(3) Proof-guided reasoning.* Finally, we fine-tune a REASONER LLM with Chain-of-Thought (CoT) supervision derived from proofs. This improves robustness to noisy retrievals and distills the structural knowledge of pre-trained link predictors into the LLM. The REASONER can be seen as performing graph traversal in the language space, with the next node determined by token probabilities.

The overall PARR framework is illustrated in Figure 1. Through extensive experiments, we show PARR achieves SOTA H@1 performance across all tested datasets, under both transductive and inductive settings. To our knowledge, this is the first native, fully generative LLM-based KGC framework to match the accuracy of discriminative link predictors.

The rest of the paper is organized as follows: Section 3 introduces background on link prediction and proof extraction. Section 4 presents the PARR framework. Section 5.2 reports experimental results across benchmark datasets and settings. Section 5.3 provides detailed ablation studies on each component of our framework. More results and ablations are provided in the appendix.

## 2 RELATED WORK

### 2.1 LINK PREDICTION ON KNOWLEDGE GRAPHS

Knowledge graph completion (KGC) is a long-standing task that seeks to infer missing links between entities. Knowledge Graph Embedding (KGE) (Bordes et al., 2013; Trouillon et al., 2016; Balazevic

et al., 2019) is an effective method for KGC. These models embed entities and relations into continuous spaces, but typically operate as black-box functions without interpretability.

**Interpretable link prediction.** Another line of works explored KG reasoning via GNN-based or rule-based methods. Notably, NBFNet (Zhu et al., 2021) solves link prediction as a shortest path finding problem and parametrize Bellman-Ford algorithm with GNN. A*Net (Zhu et al., 2023) improves the scalability of NBFNet by incorporating A* algorithm to select important nodes and edges. Neural Theorem Provers (Rocktäschel & Riedel, 2017; Cui et al., 2025) are rule-based neuro-symbolic methods that extends backward chaining algorithm into continuous space.

**Link prediction with Language Models** Leveraging pre-trained language models for link predictions has been explored for years. Earlier works leverage BERT as GNN encoder to encode triplets (Guo et al., 2022; Chen et al., 2021), while others use BERT to encode textual information of KG entities and relations (Yao et al., 2019; Wang et al., 2021; Lin et al., 2023; Youn & Tagkopoulos, 2023). These works mostly follow the same training regime used in KGE methods.

Recent works have explored LLMs for link prediction through fine-tuning (Yao et al., 2025; Zhu et al., 2024), but their performance often lags behind traditional KGE methods. On the other hand, KICGPT (Wei et al., 2023) achieves strong results by using a KGE model to generate candidate entities and asking an LLM to rerank them. However, it depends on proprietary models (ChatGPT), external KGE systems, and costly multi-round QA. MKGL (Guo et al., 2024) modifies the embedding and output spaces of LLMs to act as specialized link predictors. While effective, this approach reduces the LLM to a task-specific discriminative model, sacrificing its generative capacity.

In contrast, PARR is a fully generative, LLM-native framework that preserves the conversational ability of LLMs while achieving competitive link prediction accuracy. This generality also makes it easily adaptable to broader KG tasks such as node classification or complex KG-based QA.

## 2.2 RETRIEVAL-AUGMENTED KG REASONING

**Retrieval-Augmented Generation.** Retrieval-augmented generation (RAG) (Lewis et al., 2021) equips LLMs with external memory by retrieving relevant facts to support reasoning in knowledge-intensive tasks. Works focus on jointly pre-training or fine-tuning LLMs with retrieval modules to boost accuracy (Guu et al., 2020; Izacard & Grave, 2021; Borgeaud et al., 2022; Izacard et al., 2022).

**Query rewriting for RAG.** Query rewriting improves retrieval by transforming complex queries into simpler sub-queries, or to incorporate more information to bridge retrieval asymmetry. HyDE (Gao et al., 2022) uses LLMs to generate synthetic document for document retrieval. Ma et al. (2023); Mao et al. (2024) train query rewriters on top of retrieve-then-read systems with preference fine-tuning, while LeRet (Hsu et al., 2025) introduces an iterative rewriting strategy for multi-hop QA.

**Retrieval-Augmented KGQA.** A related line of work comes from knowledge graph question answering (KGQA), where models answer complex natural-language questions over a KG. (Das et al., 2022) retrieves nearest-neighbor subgraphs and transfers their latent reasoning patterns to new questions via a GNN, enabling efficient multi-hop reasoning without explicit supervision. RoG (Luo et al., 2024) supervises its planning and reasoning modules using retrieved paths, while GNN-RAG (Mavromatis & Karypis, 2024) uses a GNN retriever to gather subgraphs and verbalized paths as evidence for an LLM. Although these approaches share a retrieve–reason intuition, they rely on supervision such as complete paths and ground-truth retrieval, that are not available in KGC datasets. In contrast, PARR obtains the necessary structural supervision from interpretable link predictors.

## 3 PRELIMINARIES

### 3.1 KNOWLEDGE GRAPH AND LINK PREDICTION

**Knowledge Graph.** A Knowledge Graphs (KG) is a directed, multi-relation graph expressed as a set of triplets $(s, r, o) \subseteq \mathcal{E} \times \mathcal{R} \times \mathcal{E}$, where $\mathcal{E}$ and $\mathcal{R}$ denote the set of entities and relations in the KG. Link prediction is one of the most important task in the domain of KG, whose goal is to predict missing object(subject) given subject(object) entity and relation, i.e. $(s, r, ?)$ or $(?, r, o)$.

**Proofs.** For simple link prediction tasks, given any pair of entities $(s, o)$, its proof $p(s, o)$ can be defined as chain-like logical rules in the form of: $p(s, o) \leftarrow r_1(s, z_1) \wedge r_2(z_1, z_2) \wedge \cdots \wedge r_n(z_n, o)$, where $r_1, \cdots r_n$ are relations in the given KG. For path-based link predictors, the proofs can be reformulated as the paths take to reach object $o$ from subject $s$, i.e. $p(s, o) = r_1(s, z_1) \rightarrow r_2(z_1, z_2) \rightarrow \cdots \rightarrow r_n(z_n, o)$.

### 3.2 Proof extractions with Interpretable Link Predictors

We consider two SOTA link prediction models with interpretable path formulation: NBFNet (Zhu et al., 2021) and A*Net (Zhu et al., 2023). Below we briefly describe how proofs can be extracted.

**NBFNet** parametrize Bellman-Ford path-finding algorithm with GNN. Given a triplet $(s, r, o)$, we can sample top-$k$ proofs $P_1, P_2, \cdots, P_k$ from a pre-trained NBFNet through a linear model( (Baehrens et al., 2009)) (*i.e.* 1st order Taylor expansion), where each proof's score can be modeled by the partial derivative of the prediction $p(s, r, o)$ *w.r.t.* the proof path (Zhu et al., 2021):

$$P_1, P_2, \cdots, P_k = \underset{P \in \mathcal{P}_{(s,o)}}{\text{top-k}} \frac{\partial p(s, r, o)}{P}$$

In practice we use the average edge importance to approximate path score, which can be efficiently computed by auto differentiation.

**A*Net** learns a neural priority function to select a subset of nodes and edges on top of NBFNet to improve scalability. The A* algorithm provides a natural and distinct way of extracting proofs (Zhu et al., 2023). Given a triplet $(s, r, o)$ and the pre-trained node priority function $s_{(s,r)}^t(x)$, we can sample top-$k$ proofs by their average node importance $s(P)$:

$$s(P) = \frac{1}{|P|} \sum_{t=1, P^{(t)}=(x,r,y)}^{|P|} \frac{s_{s,r}^{(t-1)}(x)}{S_{s,r}^{(t-1)}}, \quad S_{s,r}^{(t-1)} = \max_{x \in \mathcal{E}^{(t-1)}} s_{s,r}^{(t-1)}(x).$$

## 4 Proof-Augmented Retrieval and Reasoning

We introduce Proof-Augmented Retrieval and Reasoning (PARR), a native LLM-based framework for link prediction. PARR is composed of three modules: a REWRITER, a REASONER, and a non-trainable RETRIEVER. We leverage proofs extracted from pre-trained interpretable link predictors to supervise the REWRITER LLM for KG query rewriting, and the REASONER LLM for structure-aware reasoning given large retrieved context. Figure 1 provides an overview of our framework.

### 4.1 Proof Extraction

We start off by extracting top-$k$ proofs for each triplet using pre-trained link predictors. To avoid model bias from the pre-trained link predictor, we sample proofs using a mixture of experts (MoE) of different models (NBFNet and A*Net), where each model has a distinct method on proof extraction 3. We further train each link predictors several times with different configurations and random seeds, and aggregate the resulting proofs. Finally, we filter out proofs with duplications or cycles. For triplets in the training set, we also filter out proofs containing the triplet itself, as it represents an existing edge for which the model is likely to give the highest score. The resulting RAG database $\mathcal{D}$ is then a set of triplet-proofs pairs: $\mathcal{D} = \left\{ \left( (s, r, o), \mathcal{P}_{s,r} \right) \right\}$, where $\mathcal{P}_{s,r} = \{p_1, \cdots, p_k\}$ denotes a set of proofs that deduce the triplet. For visualized dataset samples, please refer to Appendix E.

### 4.2 Retriever

We consider three types of fixed, non-trainable retrieval interfaces. **(1) Sparse retrieval.** Similar to KICGPT, we consider triplets that share the same subject/object with the query. To constrain retrieval size, we only select top-$k$ triplets based on the degree of their entities. **(2) Dense retrieval.** We encode triplets with text embedding models. We consider two embedding approaches: separate entity/relation embedding followed by principal neighborhood aggregation, and sentence embedding by converting triplets to natural language sentences. **(3) Hybrid retrieval.** We first perform dense retrieval, then run breadth-first-search with a predefined depth on each of retrieved proofs.

## 4.3 REWRITER

**Problem Statement.** To improve KG retrieval, we aim to train a REWRITER LLM to decompose the original query $(s, r)$ into a set of sub-queries whose individual retrievals collectively cover the triplets in ground-truth proof. Formally, let $f_\theta$ be the REWRITER that maps the query $(s, r)$ into a set of rewritten triplets:

$$\mathcal{Q} = f_\theta(s, r) = \left\{ (s^{(1)}, r^{(1)}, o^{(1)}), \cdots, (s^{(m)}, r^{(m)}, o^{(m)}) \right\}.$$

For each rewrited triplet $q_i = (s^{(i)}, r^{(i)}, o^{(i)}) \in \mathcal{Q}$, we retrieve the top-$k$ most similar KG triplets using the fixed retrieval interface detailed in Sec. 4.2, denoted as $\texttt{Retrieve}(\cdot, \cdot)$:

$$\mathcal{G} = \bigcup_{i=1}^{m} \mathcal{G}_i, \ \ \mathcal{G}_i = \texttt{Retrieve}(q_i, k).$$

We say the rewriting $\mathcal{Q}$ successfully covers the ground-truth if there exists *at least* one proof path $p \in \mathcal{P}$ such that $p \subseteq \mathcal{G}$. That is, all triplets in some valid proofs are retrieved by the composed sub-queries $\mathcal{G}$. This forms our retrieval success criterion.

**Constructing minimal rewriting sets.** To generate rewriting data for finetuning the REWRITER, we extract Minimal Rewriting Sets (MRS) that satisfy the proof coverage constraint discussed above. The extraction algorithm is provided in Appendix 1. For each proof path $p_j \in \mathcal{P}$, we solve a set-cover problem to find a minimal set of rewritten queries $\mathcal{Q}$ such that:

$$p_j \subseteq \text{argmin}_{|\mathcal{Q}|} \bigcup_{q \in \mathcal{Q}} \texttt{Retrieve}(q, k).$$

Since there may exist multiple possible rewriting combinations that fully cover a given proof path, we run a beam-search style iteration on top of the set-cover algorithm to obtain top-$n$ MRS $\mathfrak{Q}^*$ for each proof. We are therefore able to obtain a much larger training dataset (as compared to the original training set size for link prediction) for REWRITER of size $|N| \cdot |\mathcal{P}| \cdot |\mathfrak{Q}^*|$, where $|N|$ is the number of training triplets, $|\mathcal{P}|$ is the number of proofs per triplet, and $|\mathfrak{Q}^*|$ is the number of minimal rewriting sets per proof. The resulting dataset is composed by query-rewriting pairs $\big((s, r), \mathcal{Q}\big)$. This expanded training set allows the REWRITER to learn a more comprehensive mapping between the query and the rewriting sets. The detailed statistics for the resulting dataset can be found in Appendix F.

**Finetuning REWRITER.** Given a query $(s, r)$, we want to model the space of all valid query decompositions that lead to a successful coverage. Formally, let $\mathfrak{Q}^*$ denote the set of all valid rewrite sets $\mathcal{Q} = \{q_1, \cdots, q_m\}$ such that the retrieval result $\mathcal{G}(\mathcal{Q})$ covers at least one proof path in $\mathcal{P}_{s,r}$, that is $\mathfrak{Q}^* = \big\{ \mathcal{Q} : \exists p \in \mathcal{P} \text{ s.t. } p \subseteq \mathcal{G}(\mathcal{Q}) \big\}$. We want to maximize the conditional distribution:

$$\mathcal{L}_f(\theta) = -\log \sum_{\mathcal{Q} \in \mathfrak{Q}^*} f_\theta(\mathcal{Q}|s, r).$$

which is optimized with standard negative log-likelihood (NLL) loss in LLM finetuning. Sample prompt and LLM output can be found in Appendix F.

## 4.4 REASONER

While modern LLMs are trained on vast amounts of reasoning data, they remain ineffective for link prediction due to two factors: (1) limited robustness when retrievals contain irrelevant or noisy context (Fig. 3), and (2) insufficient understanding of the structural and relational patterns in knowledge graphs. To address both issues, we fine-tune the REASONER with Chain-of-Thought (CoT) supervision using retrieved context and extracted proofs.

**Problem Statement.** Given a link prediction query $(s, r)$, and the retrieved set $\mathcal{G}$, REASONER $g_\phi$ predicts the missing object $o$ after producing an intermediate CoT:

$$g_\phi : \big((s, r), \mathcal{G}\big) \mapsto (p, o), p = [(s, r^{(1)}, o^{(1)}), (o^{(1)}, r^{(2)}, o^{(2)}), \cdots, (o^{(n-1)}, r^{(n)}, o)].$$

During inference, the finetuned REASONER is essentially performing graph traversal over the subgraph provided by the RETRIEVER, by predicting the next most probable token (node).

**Constructing reasoning dataset.** Since each fact can be proved in multiple ways, and each proof can be supported by different rewriting sets, we expand the training data analogously to the rewriting dataset (Sec. 4.3). The resulting dataset consists of quadruplets $((s, r), \mathcal{G}, p, o)$, with size $|N| \cdot |\mathcal{P}| \cdot |\mathfrak{Q}^*|$, matching that of the rewriting training set. Since retrieval is often imperfect, we randomly drop a portion of ground-truth retrievals during REASONER fine-tuning. This encourages the model to infer missing facts from incomplete context, improving robustness to noisy or partial retrievals.

### 4.5 EXTENDING TO MULTI-ANSWER LINK PREDICTION

The framework described so far addresses single-answer link prediction, where the model is trained to retrieve and reason towards a single correct object $o$ for a query $(s, r)$. To align with the standard evaluation setting, which considers top-$k$ answers (e.g., $k = 10$), we extend both retrieval and reasoning to support multi-answer supervision. This contrasts with prior LLM-based approaches (Yao et al., 2025), which are restricted to top-1 prediction.

**Sampling top-$k$ answers and proofs.** Since for each $(s, r)$ pair, there may be less than $k$ ground-truth tail entity $t$, we use the score distribution from interpretable link predictors (the same models that generate proofs) to obtain additional candidate $t$s with scores above a preset threshold. Specifically, given a query $(s, r)$, the link predictor returns a ranked list $\mathcal{O}_{\text{top-}k} = o_1, \ldots, o_k$. For each candidate $o_i$, we extract an associated proof set $\mathcal{P}_i = p_1^i, \ldots, p_m^i$.

**Extending rewriter and reasoner.** In the multi-answer setting, REWRITER outputs a set of subqueries such that the resulting retrieved triplets $\mathcal{G}(Q)$ cover at least one proof path for each candidate:

$$\forall i \in \{1, \cdots, k\}, \exists p \in \mathcal{P}_i \text{ s.t. } p \subseteq \mathcal{G}(Q).$$

Similarly, the REASONER is trained not only to produce a single object but to output a ranked list of candidates, each accompanied by its proof as chain-of-thought (CoT).

## 5 EXPERIMENTS

### 5.1 EXPERIMENT SETUP

**Datasets.** We evaluate PARR on FB15k-237 (Toutanova & Chen, 2015) and WN18RR (Dettmers et al., 2018), two major datasets for link prediction. We consider both transductive and inductive setting, following standard splits in (Trouillon et al., 2016) and (Teru et al., 2020). Dataset statistics can be found in Appendix E.1.

**Evaluation.** We evaluate with the standard HITS@$k$ metrics: HITS@1, HITS@3 and HITS@10. We do not consider mean reciprocal recall (MRR) because we do not produce the full score distribution.

**Implementation Details.** We employ Llama3-8B Instruct (lla, 2024) (abbr. as Llama3) and Qwen3-8B (Yang et al., 2024) (abbr. as Qwen3) as the base LLM. We finetune using LoRA (Hu et al., 2021) with rank and alpha being 32 for one epoch. By default we use dense retrieving and sentence embedding with Jina V3 (Sturua et al., 2024) embedding model as the fixed RETRIEVER for our REWRITER model. We employ NBFNet (Zhu et al., 2021) and A*Net (Zhu et al., 2023) as the interpretable expert models. For each model, we train separately twice with different message functions and random seeds. Please refer to Appendix E.1 for full implementation details.

**Baselines.** We compare PARR against embedding based models such as TransE (Bordes et al., 2013), RotatE (Sun et al., 2019), ComplEx (Trouillon et al., 2016), and TuckER (Balazevic et al., 2019); GNN-based methods like CompGCN (Vashishth et al., 2020) and NBFNet (Zhu et al., 2021); path-based methods such including NeuralLP (Yang et al., 2017) and A*Net (Zhu et al., 2023); and methods that utilize pre-trained language models including KG-BERT (Yao et al., 2019), StAR (Wang et al., 2021), KGLM (Youn & Tagkopoulos, 2023), FTL-LM (Lin et al., 2023), DET (Guo et al., 2022), KG-Llama (Yao et al., 2025), KICGPT (Wei et al., 2023) and MKGL (Guo et al., 2024).

Table 1: Transductive link prediction on FB15K-237, WN18RR and NELL995. We use Llama3/Qwen3 to refer to Llama3-8B-instruct/Qwen3-8B throughout the rest of the paper. Best/2nd-best results are in **Bold**/underlined.

| Model | FB15K-237 | | | WN18RR | | | NELL995 | |
|---|---|---|---|---|---|---|---|---|
| | HITS@1 | HITS@3 | HITS@10 | HITS@1 | HITS@3 | HITS@10 | HITS@1 | HITS@10 |
| TransE | 0.218 | 0.345 | 0.495 | 0.061 | 0.366 | 0.522 | - | - |
| RotatE | 0.241 | 0.375 | 0.533 | 0.428 | 0.492 | 0.571 | 0.448 | 0.608 |
| ComplEx+RP | 0.298 | 0.425 | 0.568 | 0.443 | 0.505 | 0.578 | - | - |
| TuckER | 0.266 | 0.394 | 0.544 | 0.443 | 0.526 | 0.526 | - | - |
| CompGCN | 0.264 | 0.39 | 0.535 | 0.443 | 0.494 | 0.546 | 0.257 | 0.544 |
| NeuralLP | - | - | 0.362 | 0.371 | 0.434 | 0.566 | - | - |
| Red-GNN | 0.283 | - | 0.558 | 0.485 | - | 0.624 | 0.476 | 0.651 |
| NBFNet | 0.321 | 0.454 | **0.599** | 0.497 | 0.573 | 0.666 | 0.485 | 0.655 |
| A*Net | 0.321 | 0.453 | 0.586 | 0.495 | 0.573 | 0.659 | 0.479 | 0.652 |
| KG-BERT | - | - | 0.420 | 0.041 | 0.302 | 0.524 | - | - |
| StAR | 0.205 | 0.322 | 0.482 | 0.243 | 0.491 | 0.709 | - | - |
| KGLM | 0.200 | 0.314 | 0.468 | 0.330 | 0.538 | 0.741 | - | - |
| FTL-LM | 0.253 | 0.386 | 0.521 | 0.452 | 0.637 | **0.773** | - | - |
| MKGL | 0.325 | 0.454 | 0.591 | 0.500 | 0.577 | 0.656 | - | - |
| KG-Llama-7b | - | - | - | 0.242 | - | - | - | - |
| GPT 3.5 Turbo | 0.267 | - | - | 0.212 | - | - | - | - |
| KICGPT | 0.327 | 0.448 | 0.554 | 0.474 | **0.585** | 0.641 | - | - |
| PARR-Llama3 (ours) | 0.344 | 0.453 | 0.588 | 0.496 | 0.573 | 0.641 | 0.514 | 0.655 |
| PARR-Qwen3 (ours) | **0.352** | **0.465** | 0.593 | **0.513** | 0.584 | 0.653 | **0.519** | **0.658** |

Table 2: Transductive setting on YAGO3-10 dataset. Baselines are from Zhu et al. (2023).

Table 3: Performance on inductive KG reasoning on FB15k-237-ind (v1) and WN18RR-ind (v1). Baseline results are from (Zhu et al., 2021; Guo et al., 2024).

| Method | YAGO3-10 | | |
|---|---|---|---|
| | HITS@1 | HITS@3 | HITS@10 |
| DistMult | 0.24 | 0.38 | 0.54 |
| ComplEx | 0.26 | 0.40 | 0.55 |
| RotatE | 0.402 | 0.550 | 0.670 |
| BoxE | 0.400 | 0.472 | 0.541 |
| HAKE | 0.452 | 0.516 | 0.582 |
| NFBNet | 0.480 | 0.612 | **0.708** |
| A*Net | 0.470 | 0.611 | 0.707 |
| KG-LLaMA-13B | 0.133 | - | - |
| PARR-Qwen3 | **0.494** | **0.620** | 0.691 |

| Model | FB15K-237-ind | | WN18RR-ind | |
|---|---|---|---|---|
| | H@1 | H@10 | H@1 | H@10 |
| NeuralLP | 0.243 | 0.468 | 0.592 | 0.772 |
| DRUM | 0.247 | 0.474 | 0.613 | 0.777 |
| GraIL | 0.302 | 0.483 | 0.653 | 0.769 |
| RED-GNN | 0.302 | 0.483 | 0.653 | 0.8 |
| NBFNet | 0.335 | 0.574 | 0.695 | **0.826** |
| MKGL | 0.400 | **0.595** | 0.700 | 0.822 |
| ChatGPT [ 42] | 0.288 | - | 0.279 | - |
| PARR-Llama3 | 0.394 | 0.541 | 0.711 | 0.813 |
| PARR-Qwen3 | **0.412** | 0.566 | **0.718** | 0.824 |

## 5.2 MAIN RESULTS

Table 1 summarizes the results on KG link prediction under the transductive setting. PARR shows competitive performance against existing methods on both datasets. Particularly, PARR-Qwen3 outperforms previous SOTA (MKGL) on HITS@1 for both FB15K-237 and WN18RR. For instance, PARR-Qwen3 achieves 0.348 HITS@1 on FB15K-237, 2.3% above the previous SOTA (MKGL).

Table 3 shows results under the inductive setting. PARR-Qwen3 noticeably outperforms all the other methods under HITS@1 for both datasets. For example, PARR-Qwen3 scores 0.412 HITS@1 on FB15K-237-ind, surpassing previous SOTA by 1.2%. In Table 2 we show additional results on YAGO3-10 Mahdisoltani et al. (2015) with PARR-Qwen3. We can observe PARR achieves better performance on HITS@1 and HITS@3, and is comparable to SOTA methods on HITS@10.

In both transductive and inductive settings, we observe PARR achieves stronger performance on metrics with small $k$s, such as HITS@1 and HITS@3, while relatively fall short on HITS@10 as compared to SOTA methods such as NBFNet and MKGL. This is due to existing methods such as NBFNet and MKGL directly learn a score distribution over all entities, which is in turn more advantageous when considering large $k$s.

## 5.3 ABLATION STUDIES

**Evaluating Retrieval.** In Figure 2 we show average recall and different number of retrievals per sample (controlled by top-$k$ retrieval). We can see the REWRITER outperforms all other retrieving

Table 4: Retrieval performance of PARR REWRITER under different top-$k$, and their effect on the downstream REASONER's performance. The recall is calculated as the highest among $n$ ground-truth paths. top-$k$ for REWRITER refers to top-$k$ retrievals given each rewrited sub-queries. Avg. Retr. refers to average number of retrievals per sample; Avg. Rew. refers to the average number of rewrited sub-queries per sample. Avg. Retr. is caluculated from Avg. Rew. $\times$ $k$ $\times$ Path-Lengths. Best/2nd-best results are in **Bold**/underlined.

| top-$k$ | Base LLM | Avg. Retr. | Avg. Rew. | Recall($\uparrow$) | HITS@1 | HITS@3 | HITS@10 |
|---|---|---|---|---|---|---|---|
| | | | FB15K-237 | | | | |
| 2 | Llama3 | 23 | 4 | 0.452 | 0.288 | 0.288 | 0.371 |
| 2 | Qwen3 | 20 | 4 | 0.485 | 0.319 | 0.413 | 0.406 |
| 5 | Llama3 | 26 | 2 | 0.741 | 0.341 | 0.449 | 0.579 |
| 5 | Qwen3 | 25 | 2 | **0.758** | **0.348** | **0.460** | **0.586** |
| 15 | Llama3 | 48 | 1 | 0.724 | 0.312 | 0.426 | 0.534 |
| 15 | Qwen3 | 50 | 1 | 0.743 | 0.325 | 0.437 | 0.548 |
| | | | WN18RR | | | | |
| 2 | Llama3 | 27 | 5 | 0.388 | 0.251 | 0.352 | 0.325 |
| 2 | Qwen3 | 26 | 5 | 0.395 | 0.262 | 0.378 | 0.403 |
| 5 | Llama3 | 29 | 2 | 0.706 | 0.488 | 0.566 | 0.632 |
| 5 | Qwen3 | 30 | 2 | **0.713** | **0.507** | **0.581** | **0.644** |
| 15 | Llama3 | 44 | 1 | 0.662 | 0.462 | 0.514 | 0.541 |
| 15 | Qwen3 | 44 | 1 | 0.683 | 0.476 | 0.533 | 0.590 |

methods significantly in terms of Recall, while requiring much fewer number of retrievals, for both base LLMs and datasets. Meanwhile, Dense + proof paths noticeably outperform Dense without proof paths in all scenarios, demonstrating the effect of augmenting proof paths into retrieval. On the other hand, KICGPT's retrieval shows similar performance to the baseline sparse/dense RETRIEVER under our implementation, but falls significantly behind Dense + proof paths and REWRITER. In case of the non-trainable RETRIEVER, we observe dense mode to slightly outperform sparse retrieval, and hybrid RETRIEVER being marginally better than dense RETRIEVER.

**Effect of Retrieval Performance on Reasoner.** In Table 4 we show ablations on the effect of top-$k$ on our REWRITER, and the effect of the retrieval performance on the REASONER LLM. We can observe best retrieval performance are achieved at $k = 5$ under all scenarios, with $k = 2$'s recall lags significantly behind. We conjecture this is because when $k$ is low, the number of rewrited sub-queries are noticeably increased. This imposes a harder task on the REWRITER, which has to learn a much more complex mapping between the query and the rewrited sub-queries, leading to low recall. On the other hand, we can also observe the downstream REASONER's performance is directly tied to the quality of the retrieval.

Figure 2: Retrieval performance on FB15K-237 and WN18RR as a function of Average Retrievals per Sample ($x$-axis) and Average Recall($y$-axis. + proof paths denotes dense RETRIEVER with proofs included in the retrieval. + proof paths + BFS(2) refers to the hybrid RETRIEVER with Breadth-First-Search depth of 2.

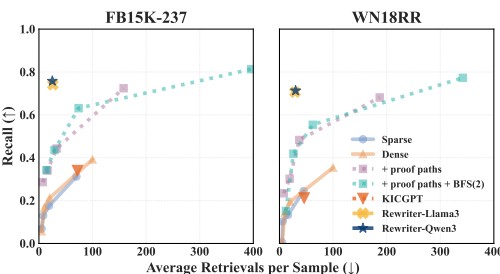

**Ablations on Reasoner.** We start by evaluating the inherent logical reasoning ability of LLMs. In Figure 3 we show zero-shot performance on link prediction, where ground-truth retrievals (grounding paths extracted from pretrained link predictors) are provided. To simulate noisy retrievals with false positives, we randomly sample and inject irrelevant triplets. We can see the model's performance lags largely behind traditional link predictors, even with ground-truth retrieval and no false positives (*i.e.* recall=precision=1). The model performance quickly degrade as the number of injected irrelevant triplets increase. For FB15K-237, both LLMs's HITS@1 quickly drop to near 0 with 16 false positives. This shows the need for a model with better and more robust logical reasoning capacity.

In Table 5 we show the impact of each component on the performance of the REASONER. We consider two LLM-based baselines: (1) LLM with naive finetuning (without CoT and retrieval). Similar to KG-Llama, we perform finetuning to predict missing entities, without retrieval. (2) LLM with RETRIEVER. In this setting we use the fixed RETRIEVER (Sec. 4.1).

Table 5: Ablations on the performance of REASONER with respect to each component: Chain of Thought (**CoT**), retrieval method (**Retr.**), and **Retr. Dropout**: whether to apply random dropout on ground-truth retrievals while finetuning the REASONER. Dense refers to fixed dense retrieval with $k = 50$. REW refers to our fine-tuned REWRITER. H@$k$ refers to HITS@$k$.

| Method | CoT | Retr. | Retr. Dropout | FB15K-237 H@1 | H@3 | H@10 | WN18RR H@1 | H@3 | H@10 |
|---|---|---|---|---|---|---|---|---|---|
| NBFNet | - | - | - | 0.321 | 0.454 | **0.599** | 0.497 | 0.573 | **0.666** |
| KICGPT | - | - | - | 0.327 | 0.448 | 0.554 | 0.474 | **0.585** | 0.641 |
| KG-Llama-7B | - | - | - | - | - | - | 0.242 | - | - |
| Llama3 | × | × | - | 0.077 | - | - | 0.171 | - | - |
| Qwen3 | × | × | - | 0.085 | - | - | 0.184 | - | - |
| Llama3 | × | Dense | - | 0.143 | - | - | 0.279 | - | - |
| Qwen3 | × | Dense | - | 0.155 | - | - | 0.285 | - | - |
| PARR-Llama3 | × | REW | × | 0.293 | 0.395 | 0.412 | 0.451 | 0.413 | 0.479 |
| PARR-Qwen3 | × | REW | × | 0.302 | 0.388 | 0.48 | 0.467 | 0.499 | 0.544 |
| PARR-Llama3 | ✓ | REW | × | 0.316 | 0.405 | 0.534 | 0.479 | 0.491 | 0.538 |
| PARR-Qwen3 | ✓ | REW | × | 0.335 | 0.413 | 0.521 | 0.493 | 0.524 | 0.603 |
| PARR-Llama3 | ✓ | REW | ✓ | 0.341 | 0.449 | 0.579 | 0.488 | 0.566 | 0.632 |
| PARR-Qwen3 | ✓ | REW | ✓ | **0.348** | **0.460** | 0.586 | **0.507** | 0.581 | 0.644 |

Table 6: Ablation on MoE effect for RETRIEVER recall and REASONER's HITS@1 on FB15k-237 and WN18RR.

| w/ MoE | Retriever (Recall) FB15k-237 | WN18RR | Reasoner (HITS@1) FB15k-237 | WN18RR |
|---|---|---|---|---|
| × | 0.683 | 0.629 | 0.324 | 0.456 |
| ✓ | 0.741 | 0.706 | 0.341 | 0.488 |

Figure 3: Zero-Shot performance on FB15k-237 and WN18RR. We provide ground-truth retrievals in the context with varying amount of randomly sampled irrelevant triplets, simulating the false positive retrievals.

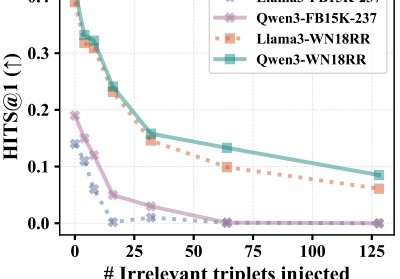

We can first observe that naive finetuning (*i.e.* directly finetuning using training triplets without retrieval or CoT) results in significantly lower accuracy in all scenarios. For instance, Llama3 only achieves 0.077 HITS@1 on FB15K-237, and 0.171 on WN18RR. With dense RETRIEVER included, the accuracy for both datasets are considerably improved, but still noticeably lag behind other models.

At the bottom of Table 5 we show each component's effect on the performance of REASONER. We observe improved performance with CoT and retrieval dropout included during training. The full PARR performs significantly better than LLM baselines, and is on par or better with existing SOTA methods.

## 5.4 ABLATION ON MoE LINK PREDICTORS

To understand the effect of sampling proofs from MoE link predictors (Sec. 4.1), we conduct ablation study on the effect of using MoE-sampled proofs versus proof sampled from a single link predictor. As shown in Tab. 6, we can observe that with MoE we can achieve notable improvement for both retrieval and link prediction accuracy, with over $6\%$ improvement on recall, and $2\%$ on HITS@1.

## 6 CONCLUSION

In this paper we introduce Proof-Augmented Retrieval and Reasoning (PARR) for KG completion. We effectively leverage proofs extracted from interpretable link predictors such as NBFNet and A*Net to (1) augment the RAG database for better sub-graph retrieval, (2) serve as golden retrieval for supervising a REWRITER LLM for query rewriting, and (3) supervise a REASONER LLM as CoT data. By experimenting on different datasets and task settings, we show PARR achieves competitive performance compared to SOTA link prediction models. Finally, we conduct extensive ablations to examine the effect and performance of each component of PARR.

ETHICS STATEMENT

We reviewed the ICLR Code of Ethics carefully and do not observe potential concerns for our work.

REPRODUCIBILITY STATEMENT

We made our best efforts to comprehensively document the implementation details. Training hyper-parameters and model architectures are discussed in Section 5.1. We include the dataset construction details including all the example prompts we used in Section E.1 and Section F.

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

Table 7: Training time comparison using default settings for each model.

| Model | WN18RR | | FB15k-237 | | YAGO | |
|---|---|---|---|---|---|---|
| | Time (hr) | Max GPU (G) | Time (hr) | Max GPU (G) | Time (hr) | Max GPU (G) |
| NBFNet | 21 | 27 | 32 | 19 | 1141 | 27 |
| AstarNet | 15 | 5 | 29 | 14 | 503 | 15 |
| MKGL (Llama2-7B) | 20 | 58 | 60 | 67 | 3789 | 54 |
| PARR (Llama3-8B) | 73 | 75 | 126 | 68 | 923 | 72 |

Table 8: Evaluation throughput (seconds per batch) and maximum GPU memory allocation (in GB).

| Model | WN18RR | | FB15k-237 | | YAGO | |
|---|---|---|---|---|---|---|
| | s/batch | Max GPU (G) | s/batch | Max GPU (G) | s/batch | Max GPU (G) |
| NBFNet | 1.3 | 1 | 1.4 | 0.6 | 3.2 | 5 |
| AstarNet | 1.3 | 0.4 | 1.4 | 0.6 | 1.7 | 4 |
| MKGL | 3.1 | 39 | 4.6 | 28 | 5.6 | 31 |
| KICGPT | 158 | – | 176 | – | – | – |
| PARR (Llama3-8B) + vLLM | 3.2 | 74 | 3.5 | 74 | 3.3 | 74 |

## A    COMPUTATION ANALYSIS

Here we profile the training/evaluation's peak GPU memory consumption and time spent for PARR and SOTA methods. All experiments are run on the same compute node with one Nvidia A100 GPU. Due to time limit, we estimate training time by average time per step $\times$ steps per epoch $\times$ n epochs using parameters from models' default configs. For evaluation, we measure the average time (in second) spent for one batch with batch size equals to 8.

**Justification on the system complexity.** While the overall PARR framework exhibits more complexity than traditional link predictors, we would like to note:

(1)    PARR targets a fundamentally different use case than traditional link predictors. While traditional link predictors such as KGE models are discriminative classifiers, PARR is built towards a generative, conversational KG agent. This means PARR must model the full distribution of the KG over natural language's space, which is substantially more challenging than training a discriminative KGE with a binary classification loss.

(2)    PARR offers significantly better inference efficiency than prompt-based LLM approaches. Unlike recent SOTA prompt-based LLM KGC method such as KICGPT (Wei et al., 2023), which require repeated calls to large proprietary LLMs, PARR is much more efficient at inference time. As shown in Table 8, on FB15k-237, KICGPT takes 176 seconds to evaluate 8 samples, whereas PARR takes only 3.5 seconds—with better accuracy.

(3)    PARR is scalable. GNN-based link predictors often incur at least polynomial (exponential for exhaustive path search algorithms like PathCon) time/memory complexity w.r.t. to the number of entities, edges, and embedding dimensions. In contrast, both REWRITER and REASONER in PARR have near constant time and memory consumption, irrespective of graph size. The RETRIEVER can also achieve constant or logarithmic time complexity via precomputed retrieval tables or fast retrieval libraries like FAISS (Johnson et al., 2019). As shown in Tab. 7, on YAGO3-10, MKGL requires ~3700 GPU hours, while PARR achieves better performance with only ~920 GPU hours.

(4)    Heavy data curation, straightforward deployment. Despite the complexity of PARR, more than half of the efforts are for data curation. This, however, has become a normal practice in the era of LLMs, where data curation has become the most important and time-consuming process. On the other hand, PARR is straightforward to deploy in real-world, thanks to the development in LLM acceleration tools (e.g. vLLM) and scalable retrieval index (e.g. FAISS).

Table 9: Training with Llama3-8B (total hours spent).

| Dataset | Rewriter | Retriever | Reasoner |
|---------|----------|-----------|----------|
| WN18RR | 15 | – | 58 |
| FB15k-237 | 23 | – | 103 |
| YAGO3-10 | 88 | – | 835 |

Table 10: Evaluation with Llama3-8B (seconds per batch, with vLLM).

| Dataset | Rewriter | Retriever | Reasoner |
|---------|----------|-----------|----------|
| WN18RR | 1.3 | – | 1.9 |
| FB15k-237 | 1.3 | – | 2.2 |
| YAGO3-10 | 1.3 | – | 2.0 |

Table 11: Performance on inductive knowledge graph reasoning. V1-v4 refer to the 4 standard splits.

| Method | v1 | | v2 | | v3 | | v4 | |
|--------|--------|---------|--------|---------|--------|---------|--------|---------|
| | HITS@1 | HITS@10 | HITS@1 | HITS@10 | HITS@1 | HITS@10 | HITS@1 | HITS@10 |
| | | | | **FB15k-237** | | | | |
| GraIL | 0.205 | 0.429 | 0.202 | 0.424 | 0.165 | 0.424 | 0.143 | 0.389 |
| NeuralLP | 0.243 | 0.468 | 0.286 | 0.586 | 0.309 | 0.571 | 0.289 | 0.593 |
| DRUM | 0.247 | 0.474 | 0.284 | 0.595 | 0.308 | 0.571 | 0.309 | 0.593 |
| NBFNet | 0.335 | 0.574 | 0.421 | 0.685 | 0.384 | 0.637 | 0.360 | 0.627 |
| RED-GNN | 0.302 | 0.483 | 0.381 | 0.629 | 0.351 | 0.603 | 0.340 | 0.621 |
| A*Net | 0.381 | 0.589 | 0.419 | 0.672 | 0.389 | 0.629 | 0.365 | **0.645** |
| MKGL | 0.400 | **0.595** | 0.417 | 0.681 | 0.392 | **0.643** | 0.374 | **0.645** |
| PARR-Qwen3 | **0.412** | 0.586 | **0.433** | **0.688** | **0.397** | 0.633 | **0.386** | 0.633 |
| | | | | **WN18RR** | | | | |
| GraIL | 0.554 | 0.760 | 0.542 | 0.776 | 0.278 | 0.409 | 0.443 | 0.687 |
| NeuralLP | 0.592 | 0.772 | 0.575 | 0.749 | 0.304 | 0.476 | 0.583 | 0.706 |
| DRUM | 0.613 | 0.777 | 0.595 | 0.747 | 0.330 | 0.477 | 0.586 | 0.702 |
| NBFNet | 0.695 | **0.826** | 0.651 | 0.798 | 0.392 | 0.568 | 0.608 | 0.694 |
| RED-GNN | 0.653 | 0.799 | 0.633 | 0.780 | 0.368 | 0.524 | 0.606 | 0.721 |
| A*Net | 0.682 | 0.810 | 0.649 | **0.803** | 0.386 | 0.544 | 0.616 | **0.743** |
| MKGL | 0.700 | 0.822 | 0.662 | 0.799 | 0.406 | 0.559 | **0.620** | 0.741 |
| PARR-Qwen3 | **0.718** | 0.824 | **0.675** | 0.788 | **0.414** | **0.572** | **0.627** | 0.725 |

## B   MORE RESULTS

In Table 11 we show full results on the inductive settings for FB15k-237 and WN18RR. Given the better performance of Qwen3 over Llama3, we only run experiments based on Qwen3. We can see PARR achieves better or comparable performance on all the splits for both FB15k-237 and WN18RR, demonstrating its strong generalizability over unseen entities.

We further evaluate PARR on ogbl-wikikg2, a large-scale knowledge graph derived from Wikidata. The results, reported in terms of MRR, are shown in Table 12. Notably, PARR attains the strongest overall performance, despite representing a lower bound on MRR (because PARR predicts only the top-10 candidate entities, any correct entity ranked outside the top-10 receives zero reciprocal rank). Still, PARR surpasses all baselines, demonstrating the effectiveness and scalability of our proof-augmented retrieval and reasoning framework on large KGs.

To further demonstrate robustness across graph sizes and relational structures, we also evaluate on three small but widely used KGs—Kinship, Nations, and UMLS. As shown in Table 13, PARR consistently outperforms all existing systems, achieving new state-of-the-art results on all datasets across both HITS@1 and HITS@10 metrics.

Table 12: MRR results on `ogbl-wikikg2`. PARR achieves the best performance despite representing a lower bound, as it only predicts top-10 entities.

| | **PARR-Qwen3** | TransE | ComplEx | RotatE | PairRE | ComplEx+RP | A*Net |
|-----|----------------|--------|---------|--------|--------|------------|-------|
| MRR | **0.7013** | 0.4256 | 0.4027 | 0.4332 | 0.5208 | 0.6392 | 0.6767 |

Table 13: Results on three small, statistical KGs: Kinship, UMLS, and Nations. PARR achieves state-of-the-art performance across all datasets.

| Method | Kinship | | UMLS | | Nations | |
|---|---|---|---|---|---|---|
| | H@1 | H@10 | H@1 | H@10 | H@1 | H@10 |
| NeuralLP | 0.475 | 0.912 | 0.643 | 0.862 | – | – |
| MINERVA | 0.605 | 0.924 | 0.728 | 0.968 | – | – |
| DRUM | 0.367 | 0.885 | 0.546 | 0.935 | – | – |
| NBFNet | 0.632 | 0.966 | 0.721 | 0.971 | 0.633 | 0.951 |
| LERP | 0.500 | 0.931 | 0.646 | 0.942 | – | – |
| **PARR-Qwen3** | **0.656** | **0.973** | **0.748** | **0.983** | **0.672** | **0.960** |

## C  MORE ANALYSIS

### C.1  ABLATION ON THE ROBUSTNESS OF PARR ON NOISY PROOFS

Here we conduct a robustness analysis on how variations in proof quality (e.g., using weaker or partial proofs) could affect end-to-end link prediction performance. To do so, we add $n\%$ of noisy proof (proofs with bottom $5\%$ path scores) to the retrieval. In Tab. 14 we show the performance (HIT@1) of Reasoner (LLama3-8B) during inference time, where the retrieval is randomly perturbed by $n\%$. We can

Table 14: Reasoner performance (HITS@1) under different perturbation ratios.

| Dataset | 0% | 5% | 10% | 25% | 50% |
|---|---|---|---|---|---|
| FB15k-237 | 0.341 | 0.340 | 0.334 | 0.325 | 0.296 |
| WN18RR | 0.488 | 0.487 | 0.482 | 0.476 | 0.436 |

see the Reasoner maintains decent performance (e.g. less than 0.015% drop) even when perturbation rate is 25% for both datasets.

### C.2  DECOUPLING PRETRAINED LINK PREDICTORS FROM INFERENCE

To further improve test-time efficiency, we evaluate a variant of PARR in which no proofs are used during retrieval (PARR-Qwen3 w/o proofs). This removes all dependence on pre-trained link predictors during inference, making PARR lighter while preserving the training benefits of proof supervision. As shown in Table 15, this variant yields only a minor drop in performance on both FB15k-237 and WN18RR, demonstrating that PARR remains highly effective even without proof-based retrieval at test time.

Table 15: Ablation on test-time proof usage. Removing proofs during retrieval (PARR-Qwen3 w/o proofs) leads to only minimal degradation.

| Method | FB15k-237 | WN18RR |
|---|---|---|
| ComplEx+RP | 0.298 | 0.443 |
| NBFNet | 0.321 | 0.497 |
| MKGL | 0.325 | 0.500 |
| PARR-Qwen3 | **0.352** | **0.513** |
| **PARR-Qwen3 (w/o proofs)** | 0.346 | 0.508 |

### C.3  CONVERGENCE OF PARR

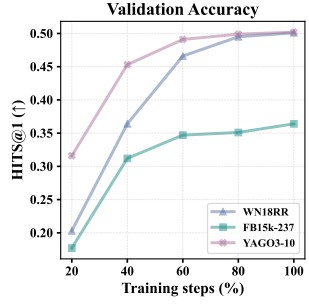

Figure 4: Validation accuracy (HITS@1) on 100 samples sampled from the validation set.

As shown in Figure 4, we observe that the PARR reasoner converges rapidly, particularly on larger KGs such as YAGO3-10, where substantial overlap arises between the constructed rewriting and reasoning sets. This property enables an efficient trade-off between training time and final performance: the model can be trained for significantly fewer steps with only minimal degradation.

# D Constructing Minimal Rewriting Sets

Given a proof and a fixed retrieval interface RETRIEVER, we want to find a minimal set of rewrited sub-queries that, after each individual retrieval, can cover the proof. We solve this with a set-cover algorithm. Moreover, since there may exist multiple (semi-)minimal rewriting sets that cover a proof, we apply a beam-search style iteration on top of the set-cover algorithm. Algorithm 1 shows the detailed procedure.

---

**Algorithm 1** Find N Minimal Rewriting Sets

---

**Require:** Target triplet $\mathcal{T}$; target proof $p$; precomputed retrievals mapping given triplets: $\mathcal{M}$; inverse of retrieval mapping $\mathcal{M}^{-1}$; maximum number of solutions $n$; beam width $w$
**Ensure:** A set of rewriting sets $\{\mathcal{Q}_1, \ldots, \mathcal{Q}_n\}$, each of which covers $p$
1: Initialize beam list: $\mathcal{B} \leftarrow \{(\emptyset, p)\}$
2: Initialize solution set: $\mathcal{S} \leftarrow []$
3: Initialize seen set: $s \leftarrow \emptyset$
4: **while** $\mathcal{B} \neq \emptyset$ **and** $|\mathcal{S}| < n$ **do**
5:     $\mathcal{B} \leftarrow []$
6:     **for all** $(\mathcal{Q}, \mathcal{U}) \in \mathcal{B}$ **do**
7:         **if** $\mathcal{U} = \emptyset$ **then**
8:             $k \leftarrow \texttt{sorted}(\mathcal{Q})$
9:             **if** $k \notin s$ **then**
10:                 $s \leftarrow s \cup \{k\}$
11:                 $S \leftarrow s \cup \{\mathcal{Q}\}$
12:             **end if**
13:             **continue**
14:         **end if**
15:         $r \leftarrow \emptyset$         ▷ Gather relevant triplets $r$ that will cover part of the remaining proof
16:         **for all** $q \in \mathcal{U}$ **do**
17:             $r \leftarrow r \cup \mathcal{M}^{-1}[q]$
18:         **end for**
19:         $r \leftarrow r \setminus \mathcal{Q} \setminus \{\mathcal{T}\}$
20:         $\texttt{Scored} \leftarrow []$   ▷ Greedily score candidate queries by how many remaining they cover
21:         **for all** $q \in r$ **do**
22:             $\mathcal{C}_q \leftarrow \mathcal{M}[q] \cap \mathcal{U}$
23:             **if** $\mathcal{C}_q \neq \emptyset$ **then**
24:                 Append $(|\mathcal{C}_q|, q, \mathcal{C}_q)$ to $\texttt{Scored}$
25:             **end if**
26:         **end for**
27:         $\texttt{Scored} \leftarrow \texttt{sorted}(\texttt{Scored}, \text{reverse=True})$
28:         **for** $i = 1$ **to** $\min(w, |\texttt{Scored}|)$ **do**
29:             $(\_, q, \mathcal{C}_q) \leftarrow \texttt{Scored}[i]$
30:             $\mathcal{Q}_{\text{new}} \leftarrow \mathcal{Q} \cup \{q\}$         ▷ Extend query set
31:             $\mathcal{U}_{\text{new}} \leftarrow \mathcal{U} \setminus \mathcal{C}_q$
32:             Append $(\mathcal{Q}_{\text{new}}, \mathcal{U}_{\text{new}})$ to $\mathcal{B}$
33:         **end for**
34:     **end for**
35:     $\mathcal{B} \leftarrow \mathcal{B}$
36: **end while**
37: **return** $\mathcal{S}$

---

# E Datasets and Implementations details

## E.1 Dataset statistics

In Table 16 and 17 we show dataset statistics for the transductive and inductive setting. We follow the standard splits in their original works.

Table 16: Dataset statistics for the transductive setting. # Rewriter Train denotes the number of training samples used for training REWRITER (and the REASONER).

| Dataset | #Entity | #Relation | # Train | # Validation | # Test | # Rewriter Train | Avg. Rewrites |
|---|---|---|---|---|---|---|---|
| FB15k-237 | 14,541 | 237 | 272,115 | 17,535 | 20,466 | 3,682,046 | 4 |
| WN18RR | 40,943 | 11 | 86,835 | 3,034 | 3,134 | 1,149,304 | 2 |
| YAGO3-10 | 123,182 | 37 | 1,079,040 | 5000 | 5000 | 16,628,192 | 5 |

Table 17: Dataset statistics for the inductive setting.

| Dataset | # Relation | Train | | Valid | | | Test | | |
|---|---|---|---|---|---|---|---|---|---|
| | | # Entity | # Triplet | # Entity | # Evaluation | # Fact | # Entity | # Evaluation | # Fact |
| FB15k-237-ind-v1 | 180 | 1,594 | 4,245 | 1,594 | 489 | 4,245 | 1,093 | 205 | 1,993 |
| FB15k-237-ind-v2 | 200 | 2,608 | 9,739 | 2,608 | 1,166 | 9,739 | 1,660 | 478 | 4,145 |
| FB15k-237-ind-v3 | 215 | 3,668 | 17,986 | 3,668 | 2,194 | 17,986 | 2,501 | 865 | 7,406 |
| FB15k-237-ind-v4 | 219 | 4,707 | 27,203 | 4,707 | 3,352 | 27,203 | 3,051 | 1,424 | 11,714 |
| WN18RR-ind-v1 | 9 | 2,746 | 5,410 | 2,746 | 630 | 5,410 | 922 | 188 | 1,618 |
| WN18RR-ind-v2 | 10 | 6,954 | 15,262 | 6,954 | 1,838 | 15,262 | 2,757 | 441 | 4,011 |
| WN18RR-ind-v3 | 11 | 12,078 | 25,901 | 12,078 | 3,097 | 25,901 | 5,084 | 605 | 6,327 |
| WN18RR-ind-v4 | 9 | 3,861 | 7,940 | 3,861 | 934 | 7,940 | 7,084 | 1,429 | 12,334 |

Table 18: Dataset samples of the rewriting training dataset for WN18RR. Entity in red refers to the target entity, which is missing from the query.

| Query | Rewrited sub-queries |
|---|---|
| *(subdivision pinophytina, hypernym, class)* | *(order coniferales, (inverse) member meronym, subdivision coniferophytina), (subdivision coniferophytina, hypernym, class)* |
| *(nippon, has part, nagasaki)* | *(volcano islands, instance hypernym, archipelago), (kyushu, has part, nagasaki), (pacific ocean, has part, volcano islands)* |
| *(enfeeble, hypernym, weaken)* | *(weaken, (inverse) hypernym, nullify), (enfeeble, drf, exhaustion)* |

Table 19: Dataset samples of the REASONERs training dataset for WN18RR. *drf* refers to *Derivationally Related Form*.

| Query | Retrievals | Target | Proof |
|---|---|---|---|
| *(screen, hypernym, ?)* | *(hood, hypernym, protective covering), (motorcar, has part, hood), (plane, has part, hood) (windshield, (inverse) has part, plane), (screen, (inverse) hypernym, windshield), ..., (hood, (inverse) has part, motorcar)* | *protective covering* | *(hood, hypernym, protective covering)* → *(hood, (inverse) has part, motorcar)* → *(screen, (inverse) hypernym, windshield)* → *(windshield, (inverse) has part, plane)* → *(plane, has part, hood)* → *(motorcar, has part, hood)* |
| *(stick in, drf, ?)* | *(inclosure, (inverse) drf, shut in), (stick in, (inverse) drf, inclosure), (inclosure, drf, stick in), ..., (stick in, (inverse) drf, inset)* | *inset* | *(stick in, (inverse) drf, inclosure)* →*(inclosure, drf, stick in)* →*(stick in, (inverse) drf, inset)* |
| *(carry, verb group, ?)* | *(carry, hypernym, make up), (make up, (inverse) hypernym, carry), (carry, (inverse) verb group, carry), (right, also see, make up), ..., (make up, (inverse) also see, right)* | *carry* | *(carry, hypernym, make up)* →*(make up, (inverse) hypernym, carry)* →*(carry, (inverse) verb group, carry)* →*(right, also see, make up)* →*(make up, (inverse) also see, right)* |

Table 20: Hyper-parameters for supervised finetuning on REWRITER and REASONER, for both Llama3-8B Instruct and Qwen3-8B.

| | Dataset | LoRA Rank/Alpha | Epochs | Learning Rate | Batch Size | Gradient Accumulation Steps | Optimizer |
|---|---|---|---|---|---|---|---|
| REWRITER | FB15k-237 | 32 | 1 | $1e^{-5}$ | 8 | 2 | AdamW |
| | WN18RR | 32 | 1 | $1e^{-5}$ | 8 | 2 | AdamW |
| | YAGO3-10 | 32 | 1 | $1e^{-5}$ | 8 | 2 | AdamW |
| REASONER | FB15k-237 | 32 | 1 | $1e^{-5}$ | 2 | 8 | AdamW |
| | WN18RR | 32 | 1 | $1e^{-5}$ | 2 | 8 | AdamW |
| | YAGO3-10 | 32 | 1 | $1e^{-5}$ | 2 | 8 | AdamW |

## E.2 DATASET SAMPLES

In Table 18 and 19 we show sample data from the rewriting dataset and reasoning dataset from WN18RR Dettmers et al. (2018).

## E.3 IMPLEMENTATION DETAILS

Table 21: Hyperparameters for NBFNet and A*Net on all datasets. We use the same set of hyper-parameters for NBFNet Zhu et al. (2021) and A*Net Zhu et al. (2023), except for parameters **Priority Function**, which are only used for A*Net.

| Hyperparameter | | **FB15k-237** | | **WN18RR** | | **YAGO3-10** |
|---|---|---|---|---|---|---|
| | | transductive | inductive | transductive | inductive | transductive |
| **Message Passing** | #step ($T$) | 6 | 6 | 6 | 6 | 6 |
| | hidden dim. | 32 | 32 | 32 | 32 | 32 |
| | message | DistMult | DistMult | DistMult | DistMult | DistMult |
| | aggregation | PNA | sum | PNA | sum | PNA |
| **Priority Function** | $g(\cdot)$ #layer | 1 | 1 | 1 | 1 | 1 |
| | $f(\cdot)$ #layer | 2 | 2 | 2 | 2 | 2 |
| | hidden dim. | 64 | 64 | 64 | 64 | 64 |
| | node ratio $\alpha$ | 10% | 50% | 10% | 5% | 10% |
| | degree ratio $\beta$ | 100% | 100% | 100% | 100% | 100% |
| **Learning** | optimizer | Adam | Adam | Adam | Adam | Adam |
| | batch size | 256 | 256 | 256 | 256 | 40 |
| | learning rate | $5e^{-3}$ | $5e^{-3}$ | $5e^{-3}$ | $5e^{-3}$ | $5e^{-3}$ |
| | #epoch | 20 | 20 | 20 | 20 | 0.4 |
| | adv. temperature | 0.5 | 0.5 | 1 | 1 | 0.5 |
| | #negative | 32 | 32 | 32 | 32 | 32 |

**Proof Extraction.** We utilize NBFNet and A*Net in this work to extract proofs. Hyperparameters are shown in Table 21. By default we adopt the same hyperparameters as original works.

**RETRIEVER.** For dense and hybrid retriever, we use Jina-V3 Sturua et al. (2024) text embedding model to compute and store the normalized embeddings, and retrieve using FAISS Johnson et al. (2019) with IndexFlatIP index, which is equivalent to the cosine-similarity metric given prenormalized embeddings. For dense retrieval, we observe similar performance when using Principle Neighboring Aggregation (PNA) and sentence embedding for triplet embedding calculation. For consistency, we use the latter throughout the paper.

**Supervised Finetuning.** We conduct experiments on Llama3-8B Instruct lla (2024) and Qwen3-8B Yang et al. (2024), two state-of-the-art LLMs. We show hyper-parameters in Table 20. We use the same set of hyper-parameters between Llama3-8B and Qwen3-8B. Experiments for FB15K-237 and WN18RR are run on 4 Nvidia H100 GPUs, and experiments for YAGO3-10 are run on 8 Nvidia H100 GPUs.

```
<|begin_of_text|><|start_header_id|>system<|end_header_id|>

You are a knowledgeable assistant that performs query rewriting for retrieving
relevant context for knowledge graph completion tasks. Given a query in the
form of a partially missing triplet (subject, relation,?), you will rewrite (expand)
the query into a list of triplets. Each of the expanded triplets is then used to
retrieve relevant triplets from a database. Your goal is to provide a list of rewrited
triplets that, after retrieval, will provide complete information needed for proving
the original query, while keeping the number of rewrited triplets minimum.
<|eot_id|><|start_header_id|>user<|end_header_id|>

Query:
( veratrum, hypernym,? )

Please expand the query into a python list of triplets in the form of [(head,
relation, tail),...]. Please strictly follow the format of the output as it will be
parsed by a program.<|eot_id|><|start_header_id|>assistant<|end_header_id|>

### Answer:[('polianthes', 'hypernym', 'liliid monocot genus'), ('hellebore',
'(inverse)  member meronym','veratrum')]<|eot_id|>
```

```
<|begin_of_text|><|start_header_id|>system<|end_header_id|>

You are a knowledgeable assistant that performs query rewriting for retrieving
relevant context for knowledge graph completion tasks. Given a query in the
form of a partially missing triplet (subject, relation,?), you will rewrite (expand)
the query into a list of triplets. Each of the expanded triplets is then used to
retrieve relevant triplets from a database. Your goal is to provide a list of rewrited
triplets that, after retrieval, will provide complete information needed for proving
the original query, while keeping the number of rewrited triplets minimum.
<|eot_id|><|start_header_id|>user<|end_header_id|>

Query:
( family compositae, member meronym,? )

Please expand the query into a python list of triplets in the form of [(head,
relation, tail),...]. Please strictly follow the format of the output as it will be
parsed by a program.<|eot_id|><|start_header_id|>assistant<|end_header_id|>

### Answer:[('vegetable oyster', 'has part','vegetable oyster'), ('genus ageratum',
'(inverse)  member meronym', 'family compositae')]<|eot_id|>
```

Figure 5: Sample prompt and answer(in dark green) for Llama3-based REWRITER.

## F   PROMPT SAMPLES

In Figure 5 and 7 we show sample prompt and answer from REWRITER LLM for WN18RR. In
Figure 6 and Figure 9 we show sample inputs and generated outputs from REASONER LLMs,
respectively.

```
<|begin_of_text|><|start_header_id|>system<|end_header_id|>

You are a knowledgeable assistant that performs link prediction for knowledge
graph completion tasks. Given a query in the form of a partially missing triplet
(subject, relation, ?), and a retrieved set of relevant triplets, each with the
corresponding reasoning steps that prove the triplet using other triplets from the
database, you will first provide the reasoning step by step based on the retrieved
triplets, and provide the appropriate object entity that completes the query.
<|eot_id|><|start_header_id|>user<|end_header_id|>

Query:
( carry, verb group, ? )

Retrieved triplets:
('land reform', 'also see', 'land reform'), ('psychopathology', '(inverse)  synset
domain topic of', 'defense reaction'), ('offset', 'derivationally related form', 'make
up'), ('make up', '(inverse)  also see'), ('compensation', 'derivationally
related form', 'overcompensate'), ('carry', 'hypernym', 'make up'), ('trim',
'hypernym', 'equilibrize'), ('carry', 'verb group', 'carry'), ('make up', 'hypernym',
'equilibrize'), ('make up', '(inverse)  hypernym', 'carry'), ('equalizer', 'derivationally
related form', 'make up'), ('carry', '(inverse)  verb group', 'carry'), ('even out', 'also see',
'make up'), ('carry', '(inverse)  verb group', 'carry'), ('equalizer', 'derivationally
related form', 'make up'), ('defense reaction', '(inverse)  hypernym',
'compensation'), ('make up', '(inverse)  derivationally related form', 'equalizer'),
('overcompensation', 'synset domain topic of', 'psychopathology'),
('overcompensate', 'derivationally related form', 'overcompensation'), ('right', 'also
see', 'make up'), ('overcompensate', 'hypernym', 'make up'), ('right', 'derivationally
related form', 'recompense'), ('right', '(inverse)  also see', 'right'), ('make up',
'(inverse)  hypernym', 'overcompensate'), ('make up', '(inverse)  derivationally
related form', 'offset'), ('recompense', '(inverse)  derivationally related form',
'make up')

What is an appropriate object entity that completes the query? Please provide a
rigorous reasoning process before the final answer.<|eot_id|>
<|start_header_id|>assistant<|end_header_id|>

### Reasoning:
Based on the retrieved context, we have:
('carry', 'hypernym', 'make up') -> ('make up', '(inverse)  hypernym', 'carry') ->
('carry', '(inverse)  verb group', 'carry') -> ('right', 'also see', 'make up') -> ('make
up', '(inverse)  also see', 'right')

Therefore, we know:
('carry', 'verb group', 'carry')

### Answer:carry<|eot_id|>
```

```
<|begin_of_text|><|start_header_id|>system<|end_header_id|>

You are a knowledgeable assistant that performs link prediction for knowledge
graph completion tasks. Given a query in the form of a partially missing triplet
(subject, relation, ?), and a retrieved set of relevant triplets, each with the
corresponding reasoning steps that prove the triplet using other triplets from the
database, you will first provide the reasoning step by step based on the retrieved
triplets, and provide the appropriate object entity that completes the query.
<|eot_id|><|start_header_id|>user<|end_header_id|>

Query:
( thread, hypernym, ? )

Retrieved triplets:
('threader', 'derivationally related form', 'thread'), ('run', '(inverse)  verb group',
'thread')]
{('yarn', 'derivationally related form', 'thread'), ('thread', 'verb group', 'run'), ('run',
'also see', 'wipe'), ('run', 'verb group', 'run'), ('thread', '(inverse)  derivationally
related form', 'threader'), ('threader', 'derivationally related form', 'thread'),
('thread', '(inverse)  derivationally related form', 'yarn'), ('threader', '(inverse)
derivationally related form', 'thread'), ('thread', 'derivationally related form',
'threader'), ('run', '(inverse)  verb group', 'thread'), ('wipe', '(inverse)  also see',
'run'), ('thread', 'derivationally related form', 'yarn'), ('run', 'verb group', 'thread'),
('thread', '(inverse)  verb group', 'run'), ('run', '(inverse)  verb group', 'run')

What is an appropriate object entity that completes the query? Please provide a
rigorous reasoning process before the final answer.<|eot_id|>
<|start_header_id|>assistant<|end_header_id|>

### Reasoning:
Based on the retrieved context, we have:
('thread', '(inverse)  derivationally related form', 'threader') -> ('thread', '(inverse)
verb group', 'run') -> ('threader', 'derivationally related form', 'thread')

Therefore, we know:
('thread', 'hypernym', 'run')
</think>

### Answer:run<|eot_id|>
```

Figure 6: Sample prompt and answer(in dark green) for Llama3-based REASONER.

<|im_start|>system
You are a knowledgeable assistant that performs query rewriting for retrieving relevant context for knowledge graph completion tasks. Given a query in the form of a partially missing triplet (subject, relation, ?), you will rewrite (expand) the query into a list of triplets. Each of the expanded triplets is then used to retrieve relevant triplets from a database. Your goal is to provide a list of rewritten triplets that, after retrieval, will provide complete information needed for proving the original query, while keeping the number of rewrited triplets minimum.
<|im_endl|>
<|im_startl|>user
Query:
( capitalization, (inverse)  derivationally related form, ? )

Please expand the query into a python list of triplets in the form of [(head, relation, tail), ...]. Please strictly follow the format of the output as it will be parsed by a program.<|im_endl|>
<|im_startl|>assistant
<think>

</think>

### Answer:[('win', '(inverse)  hypernym', 'steal'), ('profits', 'derivationally related form', 'turn a profit')]<|im_endl|>

<|im_start|>system
You are a knowledgeable assistant that performs query rewriting for retrieving relevant context for knowledge graph completion tasks. Given a query in the form of a partially missing triplet (subject, relation, ?), you will rewrite (expand) the query into a list of triplets. Each of the expanded triplets is then used to retrieve relevant triplets from a database. Your goal is to provide a list of rewrited triplets that, after retrieval, will provide complete information needed for proving the original query, while keeping the number of rewrited triplets minimum.
<|im_endl|>
<|im_startl|>user
Query:
( therapeutic, (inverse)  derivationally related form, ? )

Please expand the query into a python list of triplets in the form of [(head, relation, tail), ...]. Please strictly follow the format of the output as it will be parsed by a program.<|im_endl|>
<|im_startl|>assistant
<think>

</think>

### Answer:[('medicine', 'synset domain topic of', 'practice of medicine'), ('therapeutic', 'derivationally related form', 'heal'), ('practice of medicine', 'derivationally related form', 'medical')]<|im_endl|>

Figure 7: Sample prompt and answer(in dark green) for Qwen3-based REWRITER.

<|im_start|>system
You are a knowledgeable assistant that performs link prediction for knowledge graph completion tasks. Given a query in the form of a partially missing triplet (subject, relation, ?), and a retrieved set of relevant triplets, each with the corresponding reasoning steps that prove the triplet using other triplets from the database, you will first provide the reasoning step by step based on the retrieved triplets, and provide the appropriate object entity that completes the query.
<|im_endl|>
<|im_startl|>user
Query:
( carry, verb group, ? )

Retrieved triplets:
('land reform', 'also see', 'land reform'), ('psychopathology', '(inverse)  synset domain topic of', 'defense reaction'), ('offset', 'derivationally related form', 'make up'), ('make up', '(inverse)  also see', 'even out'), ('compensation', 'derivationally related form', 'overcompensate'), ('carry', 'hypernym', 'make up'), ('trim', 'hypernym', 'equilibrize'), ('carry', 'verb group', 'carry'), ('make up', 'hypernym', 'equilibrize'), ('make up', '(inverse)  hypernym', 'carry'), ('even out', 'also see', 'make up'), ('carry', '(inverse)  verb group', 'carry'), ('equalizer', 'derivationally related form', 'make up'), ('defense reaction', '(inverse)  hypernym', 'compensation'), ('make up', '(inverse)  derivationally related form', 'equalizer'), ('overcompensation', 'synset domain topic of', 'psychopathology'), ('overcompensate', 'derivationally related form', 'overcompensation'), ('right', 'also see', 'make up'), ('overcompensate', 'hypernym', 'make up'), ('right', 'derivationally related form', 'recompense'), ('make up', '(inverse)  also see', 'right'), ('make up', '(inverse)  hypernym', 'overcompensate'), ('make up', '(inverse)  derivationally related form', 'offset'), ('recompense', '(inverse)  derivationally related form', 'make up')

What is an appropriate object entity that completes the query? Please provide a rigorous reasoning process before the final answer.<|im_endl|>
<|im_startl|>assistant
<think>
Based on the retrieved context, we have:
('carry', 'hypernym', 'make up', '(inverse)  hypernym', 'carry') -> ('carry', '(inverse)  verb group', 'carry') -> ('right', 'also see', 'make up') -> ('make up', '(inverse)  also see', 'right')

Therefore, we know:
('carry', 'verb group', 'carry')
</think>

### Answer:carry<|im_endl|>

<|im_start|>system
You are a knowledgeable assistant that performs link prediction for knowledge graph completion tasks. Given a query in the form of a partially missing triplet (subject, relation, ?), and a retrieved set of relevant triplets, each with the corresponding reasoning steps that prove the triplet using other triplets from the database, you will first provide the reasoning step by step based on the retrieved triplets, and provide the appropriate object entity that completes the query.
<|im_endl|>
<|im_startl|>user
Query:
( thread, hypernym, ? )

Retrieved triplets:
('threader', 'derivationally related form', 'thread'), ('run', '(inverse)  verb group', 'thread')]
{('yarn', 'derivationally related form', 'thread'), ('thread', 'verb group', 'run'), ('run', 'also see', 'wipe'), ('run', 'verb group', 'run'), ('thread', '(inverse)  derivationally related form', 'threader'), ('threader', 'derivationally related form', 'thread'), ('thread', '(inverse)  derivationally related form', 'yarn'), ('threader', '(inverse)  derivationally related form', 'thread'), ('thread', 'derivationally related form', 'threader'), ('run', '(inverse)  verb group', 'thread'), ('wipe', '(inverse)  also see', 'run'), ('thread', 'derivationally related form', 'yarn'), ('run', 'verb group', 'thread'), ('thread', '(inverse)  verb group', 'run'), ('run', '(inverse)  verb group', 'run')

What is an appropriate object entity that completes the query? Please provide a rigorous reasoning process before the final answer.<|im_endl|>
<|im_startl|>assistant
<think>
Based on the retrieved context, we have:
('thread', '(inverse)  derivationally related form', 'threader') -> ('thread', '(inverse)  verb group', 'run') -> ('threader', 'derivationally related form', 'thread')

Therefore, we know:
('thread', 'hypernym', 'run')
</think>

### Answer:run<|im_endl|>

Figure 8: Sample prompt and answer(in dark green) for Qwen3-based REASONER.

<|im_start|>system
You are a knowledgeable assistant that performs multi-answer link prediction for knowledge graph completion tasks. Given a query in the form of a partially missing triplet (subject, relation, ?), and a retrieved set of relevant triplets, each with the corresponding reasoning steps that prove the triplet using other triplets from the database, you will first provide top-10 most probable reasoning step by step based on the retrieved triplets, and provide the corresponding top-10 appropriate object entity that completes the query.<|im_endl|>
<|im_startl|>user
Query:
( carry, verb group, ? )

Retrieved triplets:
('land reform', 'also see', 'land reform'), ('psychopathology', '(inverse) synset domain topic of', 'defense reaction'), ('offset', 'derivationally related form', 'make up'), ('make up', '(inverse) also see', 'even out'), ('compensation', 'derivationally related form', 'overcompensate'), ('carry', 'hypernym', 'make up'), ('trim', 'hypernym', 'equilibrize'), ('carry', 'verb group', 'carry'), ('make up', 'hypernym', 'equilibrize'), ('make up', '(inverse) hypernym', 'carry'), ('even out', 'also see', 'make up'), ('carry', '(inverse) verb group', 'carry'), ('equalizer', 'derivationally related form', 'make up'), ('defense reaction', '(inverse) hypernym', 'compensation'), ('make up', '(inverse) derivationally related form', 'equalizer'), ('overcompensation', 'synset domain topic of', 'psychopathology'), ('overcompensate', 'derivationally related form', 'overcompensation'), ('right', 'also see', 'make up'), ('overcompensate', 'hypernym', 'make up'), ('right', 'derivationally related form', 'recompense'), ('make up', '(inverse) also see', 'right'), ('make up', '(inverse) hypernym', 'overcompensate'), ('make up', '(inverse) derivationally related form', 'offset'), ('recompense', '(inverse) derivationally related form', 'make up'), ...

What are the top-10 appropriate object entities that completes the query? Please provide a rigorous reasoning process for each before the final answer.<|im_endl|>
<|im_startl|>assistant
<think>
Based on the retrieved context, we can deduce below top-10 proofs:
Proof 1:('carry', 'hypernym', 'make up') -> ('make up', '(inverse) hypernym', 'carry') -> ('carry', '(inverse) verb group', 'carry') -> ('right', 'also see', 'make up') -> ('make up', '(inverse) also see', 'right')
Proof 2: ...

Therefore, we know:
Answer 1: ('carry', 'verb group', 'carry')
Answer 2: ...
...
</think>

### Answer:[carry, ...]<|im_endl|>

<|im_start|>system
You are a knowledgeable assistant that performs multi-answer link prediction for knowledge graph completion tasks. Given a query in the form of a partially missing triplet (subject, relation, ?), and a retrieved set of relevant triplets, each with the corresponding reasoning steps that prove the triplet using other triplets from the database, you will first provide top-10 most probable reasoning step by step based on the retrieved triplets, and provide the corresponding top-10 appropriate object entity that completes the query.<|im_endl|>
<|im_startl|>user
Query:
( thread, hypernym, ? )

Retrieved triplets:
('threader', 'derivationally related form', 'thread'), ('run', '(inverse) verb group', 'thread')]
{('yarn', 'derivationally related form', 'thread'), ('thread', 'verb group', 'run'), ('run', 'also see', 'wipe'), ('run', 'verb group', 'run'), ('thread', '(inverse) derivationally related form', 'threader'), ('threader', 'derivationally related form', 'thread'), ('thread', '(inverse) derivationally related form', 'yarn'), ('threader', '(inverse) derivationally related form', 'thread'), ('thread', 'derivationally related form', 'threader'), ('run', '(inverse) verb group', 'thread'), ('wipe', '(inverse) also see', 'run'), ('thread', 'derivationally related form', 'yarn'), ('run', 'verb group', 'thread'), ('thread', '(inverse) verb group', 'run'), ('run', '(inverse) verb group', 'run'),...

What is an appropriate object entity that completes the query? Please provide a rigorous reasoning process before the final answer.<|im_endl|>
<|im_startl|>assistant
<think>
Based on the retrieved context, we have:
Proof 1: ('thread', '(inverse) derivationally related form', 'threader') -> ('thread', '(inverse) verb group', 'run') -> ('threader', 'derivationally related form', 'thread')
Proof 2: ...

Therefore, we know:
Answer 1: ('thread', 'hypernym', 'run')
Answer 2: ...
</think>

### Answer:[run,...]<|im_endl|>

Figure 9: Sample prompt and answer(in dark green) for Qwen3-based REASONER with multi-answer extension.

