# OpenReview forum: "Proof-Augmented Retrieval and Reasoning: Supervising Language models for Knowledge Graph Completion with Link Predictors"
_ICLR.cc/2026/Conference — Submitted to ICLR 2026_

### Official Review · Reviewer_cL8A · 2025-10-21

**Soundness:** 3
**Presentation:** 3
**Contribution:** 2
**Rating:** 6
**Confidence:** 5

**Summary:**

This paper proposes Proof-Augmented Retrieval and Reasoning (PARR), a general framework for finetuning language models for knowledge graph completion. PARR consists of three components, a rewriter LLM, a fixed sparse or dense retriever, and a reasoner LLM. The rewriter augments the query into several triplets so that the retriever can better recall necessary paths for proving the target answer. The reasoner then performs reasoning on the set of retrieved triplets and decide the final top-k answers. To finetune the rewriter, the authors dump path proofs from existing link predictors, namely NBFNet and A\*Net, and use them to compute several minimal sets of rewritten queries as the training signal for rewriter. The reasoner is then finetuned on the output of the rewriter and the retriever to predict the final answer. For top-k supervision, the authors distill the output from NBFNet or A\*Net to the reasoner LLM. PARR achieves state-of-the-art performance on three transductive and two inductive knowledge graph datasets.

**Strengths:**

1. Instead of using LLMs to encode text features or triplets in a KG, this paper proposes a novel way to use LLMs as a reasoner over a retrieved subgraph. This potentially leverages the rule knowledge in LLMs and their ability to perform logical deduction.
2. The framework aligns well with classical logical reasoning algorithms, where the rewriter, retriever and the reasoner correspond to backward chaining, unification and forward chaining respectively. The rewriter is responsible for recall while the reasoner is responsible for precision.
3. PARR achieves state-of-the-art performance on all popular KGC benchmarks. The prediction steps of PARR are interpretable to human.

**Weaknesses:**

1. PARR heavily relies on paths and predictions generated by NBFNet or its derivatives. To train PARR, it needs to first train an NBFNet and dump its path proofs for every triplet in the training set, which is very time consuming. If proofs are used in retrieval, one also needs to conduct NBFNet inference on the inductive test sets before applying PARR. As shown in Table 9, it takes 835 GPU hours (4 days on 8 H100) to finetune the reasoner LLM on YAGO3-10. This is way more costly than NBFNet or A*Net. For example, A*Net reports 20.8 min / epoch for YAGO3-10, which translates to 14 GPU hours. It’s not practical to spend 60x training time just for 3% gain in HITS@1.
2. Since PARR requires finetuning two LLMs, it’s not clear to me whether PARR benefits from generative LLMs or just the curated training signals. For example, we may train a small Transformer model from scratch for query rewriting (similar to the LSTM used in [1]) using the same training signal. We may train a subgraph reasoner from scratch like NBFNet. Both models only require relation embeddings and don’t need language prior.

[1] Yang et al. Differentiable Learning of Logical Rules for Knowledge Base Reasoning. NIPS 2017.

**Questions:**

1. I recommend the authors to change the term proofs to paths, or more specifically, grounding paths, for consistency with terminology in KG literature [2][3].
2. Does the initial rewriter need to be reasonably working? Otherwise, you can’t get enough minimal rewriting sets for training, right?
3. Line 261: If LLMs lack sufficient understanding of the patterns in KGs, why not train the reasoner from scratch? That would be much cheaper than a LLM.
4. Table 4: How is average number of retrievals computed? Why is it larger than average number of rewrites times k?
5. Line 398: How do you compute recall for queries with n ground truth answers?
6. Line 422: What do you mean by ground truth retrievals? Is it the path proof output by NBFNet?
7. Missing references: RAG for KGC[4], query rewriting [5].
8. Fonts for $Q$ in Line 220 and 227 are inconsistent. You may either use ordinary and mathcal fonts, or mathcal and mathfrak fonts to distinguish set $Q$ and powerset $Q^*$ respectively.
9. Line 235: please be explicit that you try to take $argmin_{|Q|}$.
10. Typo:
    1. Line 88: absed → based
    2. Line 222: Sec 4.1 → Sec 4.2
    3. Line 248: missing {} for sets.

[2] Cohen. TensorLog: A Differentiable Deductive Database. NIPS 2016.

[3] Zhu et al. Neural Bellman-Ford Networks: A General Graph Neural Network Framework for Link Prediction. NeurIPS 2021.

[4] Das and Godbole et al. Knowledge Base Question Answering by Case-based Reasoning over Subgraphs. ICML 2022.

[5] Gao and Ma et al. Precise Zero-Shot Dense Retrieval without Relevance Labels. arXiv 2022.

---

> ### Author Response · Authors · 2025-11-28
> **Response (1/2)**
>
> Thank you for your time and the insightful comments! Below we address your concerns point by point.
>
> ## Weakness
>
> > 1.Computational cost for PARR...
>
> 1. Compared to recent methods like KICGPT and MKGL, PARR is more efficient and scalable in training and inference. Unlike KGE or GNN-based methods, where complexity increase at least linear to KG size, PARR has nearly constant inference complexity, since retrieval has a fixed budget, and the KG size does not necessarily impact reasoning length (below in 4. we show that we can decouple pretrained link predictors away from inference). For instance, during training for YAGO3-10, PARR takes 923 hours (Rewriter + Reasoner), whereas MKGL takes 3789 hours. During inference for FB15k-237, KICGPT takes 176 seconds per batch, whereas PARR takees 3.5 seconds. Please refer to Table 7 and Table 8 in Appendix A for more details.
> 2. As we mentioned at the very beginning of our introduction (2nd paragraph), we believe the bottleneck for KGC is **not** efficiency: knowledge-intensive domains like healthcare and scientific discovery require deliberate reasoning and interpretability over speed, which PARR excels at. Further more, we believe PARR has many potential future directions. For instance, one of our ongoing effort is to unify different KG tasks (e.g. node classification and CQA) all under one model using PARR in a conversational manner. Therefore, we believe the additional computational is worth it.
> 3. In terms of extracting paths for each triplet, PARR is actually much more efficient than the widely-used LLM-based synthetic data creation methods. For instance, for WN18RR, with batch size=1, it requires ~4 hours with peack GPU memory < 1G. By parallelizing the process, we can finish in less than an hour.
> 4. To further enhance test-time efficiency, we additionally tested the setting where no proofs are used during retrieval (PARR-Qwen3 w/o proofs), which allows us to decouple pre-trained link predictors entirely from inference stage of PARR. We find this setting leads to little performance degradation, as shown in the below table.
>
>     | Method                     | FB15k-237 | WN18RR |
>     |---------------------------|-----------|--------|
>     | ComplEx+RP                | 0.298     | 0.443  |
>     | NBFNet                    | 0.321     | 0.497  |
>     | MKGL                      | 0.325     | 0.500  |
>     | PARR-Qwen3                | 0.352     | 0.513  |
>     | **PARR-Qwen3 (w/o proofs)**   | 0.346     | 0.508  |
> 5. Additionally, we find that PARR reasoner converges rather quickly, and especially for larger KGs (e.g. YAGO3-10), due to the overlaps in the constructed rewriting and reasoning sets. This means we can flexibly trade training time with minimal performance degradation. We show the result in below table, as well in Figure 4 (Appendix) of the revised manuscript. For instance, for YAGO3-10, we can obtain competitive performance with ~50% of the training.
>
>     |     | 20%    | 40%    | 60%    | 80%    | 100%   |
>     |------------|-------|-------|-------|-------|-------|
>     | WN18RR     | 0.203 | 0.364 | 0.466 | 0.495 | 0.501 |
>     | FB15k-237  | 0.177 | 0.292 | 0.347 | 0.351 | 0.364 |
>     | YAGO3-10   | 0.316 | 0.463 | 0.491 | 0.499 | 0.502 |
>
> > 2.Since PARR requires finetuning two LLMs, it’s not clear to me whether PARR benefits from generative LLMs or just the curated training signals. For example, we may train a small Transformer model from scratch for query rewriting (similar to the LSTM used in [1]) using the same training signal. We may train a subgraph reasoner from scratch like NBFNet. Both models only require relation embeddings and don’t need language prior.
>
> Thank you for the insightful question! There are several future directions that motivate us to enable LLMs for KG tasks:
>
> 1. **Natural Language Capacity.** Given the natural language capacity of LLMs and the flexibility, we can unify different KG-related tasks into one foundational model under the proposed PARR framework, and allow users to query the model in a flexible, conversational and interactive manner. This is one of our ongoing effort. We believe this provides a much more user-friendly interface than traditional KGE/GNN/NeSy-based link predictors.
> 2. **Generative Capacity.** While most existing KGC methods are discriminative (e.g. trained with BCE loss), PARR is generative. This means with proper supervision, PARR has the potential to generate unseen entity, relation, or even whole triplets, sampled from its language prior. This is also one of our next steps.
>
> While these two directions are not part of PARR, we believe PARR acts as a necessary first step, by first enabling LLM to "understand" the complex structure of KGs.

---

> ### Author Response · Authors · 2025-11-28
> **Response (2/2)**
>
> ## Questions
>
> > 1.I recommend the authors to change the term proofs to paths, or more specifically, grounding paths, for consistency with terminology in KG literature [2][3].
>
> We use the term "proofs" from differentiable theorem proving aspect (e.g. NTP [1]). However, we agree with the reviewer that "grounding paths" can be more suitable here. Since we use the term 'proof' extensively in the paper, we will modify it in our final manuscript to avoid confusion to other reviewers.
>
> > 2.Does the initial rewriter need to be reasonably working? Otherwise, you can’t get enough minimal rewriting sets for training, right?
>
> We do not need the rewriter to construct Minimal Rewriting Sets (MRS). The construction process of MRS is (1) extract grounding paths and build retrieval index using a pretrained text embedding model (e.g. Jina V3), and (2) solve a greedy set-cover algorithm (Algorithm 1) to find MRS. The rewriter is not involved in the process.
>
> > 3.Line 261: If LLMs lack sufficient understanding of the patterns in KGs, why not train the reasoner from scratch? That would be much cheaper than a LLM.
>
> Please kindly refer to `Weakness 2.` for details.
>
> > 4.Table 4: How is average number of retrievals computed? Why is it larger than average number of rewrites times k?
>
> Thank you for the question! The average number of retrivals is larger than average number of rewrites times $k$ because we not only retrieve $k$ triplets, but also their corresponding grounding path. We have added an explanation in the caption of Table 4.
>
> > 5.Line 398: How do you compute recall for queries with n ground truth answers?
>
> Thank you for the question. The recall is calculated by the highest recall among $n$ ground truth paths. We have added an explanation in the caption of Table 4.
>
> > Line 422: What do you mean by ground truth retrievals? Is it the path proof output by NBFNet?
>
> Exactly! We have added an explanation to the manuscript.
>
> > Missing references: RAG for KGC[4], query rewriting [5].
>
> Thank you for mentioning these works! We have incorporate them into the related work section.
>
> > 8. Fonts for $Q$ in Line 220 and 227 are inconsistent. You may either use ordinary and mathcal fonts, or mathcal and mathfrak fonts to distinguish set $Q$ and powerset $Q^*$ respectively.
> > 9. Line 235: please be explicit that you try to take $argmin_{|Q|}$.
> > 10. Typos.
>
> Thank you for the detailed advice! We have revised the manuscript.
>
>
> [1] End-to-End Differentiable Proving

---

### Official Review · Reviewer_tbAJ · 2025-10-22

**Soundness:** 2
**Presentation:** 2
**Contribution:** 2
**Rating:** 2
**Confidence:** 3

**Summary:**

This paper proposes PARR — Proof-Augmented Retrieval and Reasoning, a native generative LLM framework for knowledge graph completion (KGC). Rather than treating an LLM as a reranker or discriminative scorer, PARR retrieves proofs from pre-trained link predictors and applies LLMs to generate answers based on retrieved proofs. Experiments across a few benchmarks show comparable performance against previous LLM-based methods.

**Strengths:**

1.	Novel Framework: The PARR framework presents a new retrieval-augmented reasoning approach for applying LLMs to knowledge graph completion, offering a fresh perspective compared to previous methods that mainly used LLMs as encoders or lacked structural reasoning as well as suffered from hallucination.
2.	Combination with Link Predictors: By leveraging pre-trained link predictors to retrieve relevant proofs, PARR effectively combines symbolic reasoning with the generative capabilities of LLMs, enhancing the overall reasoning process and reducing the LLM hallucination.
3.	Empirical Validation: The authors conduct extensive experiments on multiple benchmarks, demonstrating that PARR achieves comparable performance to existing LLM-based methods, validating the effectiveness of their proposed framework.

**Weaknesses:**

1.	Limited Novelty: While the retrieval and reasoning framework is novel for KGC, similar frameworks have been widely explored in other similar tasks, especially in knowledge graph question answering (KGQA). The KGC task itself is actually a special case of KGQA where the question is explicitly given as a triple with a missing entity.
The idea of retrieving relevant information (e.g., path-like proofs) and using LLMs to reason over them has been explored in prior works such as RoG [1]. A similar approach, utilizing GNNs to retrieve paths and then employing LLMs to generate answers, has also been explored in the KGQA literature [2].
Therefore, the novelty of the proposed framework in the context of KGC is somewhat limited.
The authors should better clarify the differences and connections between their work and prior works in KGQA, and highlight the unique contributions of PARR specifically for KGC.
2.	Unsatisfied Claims on Hallucination Reduction: The authors claim that PARR advances previous LLM-based methods by reducing hallucination in prediction through the use of retrieved proofs. However, the proposed rewriter is still purely based on LLMs' internal knowledge learned from training. Therefore, it still has the potential to generate hallucinated facts that do not exist in the knowledge graph. The authors should provide more empirical evidence to evaluate the rewrite results and support their claim on hallucination reduction.
3.	Training Complexity: The training cost of PARR is relatively high, as it involves training LLMs on a large amount of generated data. The method needs to retrieve proofs for all triples in KGs and generate corresponding reasoning data for LLM training, which can be computationally expensive and time-consuming, making it less practical for large-scale KGs.
4.	Limited Performance and Generalizability: While the proposed framework involves heavy LLMs and training, its performance and generalizability are still limited compared to traditional KGC methods like NBFNet. For example, on FB15k-237 Hits@10, NBFNet achieves 0.599 while PARR only achieves 0.593. Moreover, the experiments in inductive settings show that PARR's performance drops significantly, indicating limited generalizability to unseen KGs.
Considering the proposed method is built upon the traditional KGC methods like NBFNet, it is not convincing to me that the introduction of LLMs and the complex training process is well justified, given the limited performance gains and generalizability.
5.	Minor Issues: The multi-answer generation setting is not reasonable. Considering the original KG contains 1-to-many relations, it is not necessary to sample top-k predictions from link predictors as ground truth answers.
6.	Typos and Notations: The writing and structure of the paper can be improved. There are many short sections that can be merged for better readability and flow. There are also many notation inconsistencies, e.g., using both R to denote relations and retrieved triples.

[1] LUO, LINHAO, et al. "Reasoning on Graphs: Faithful and Interpretable Large Language Model Reasoning." The Twelfth International Conference on Learning Representations.

[2] Mavromatis, Costas, and George Karypis. "Gnn-rag: Graph neural retrieval for efficient large language model reasoning on knowledge graphs." Findings of the Association for Computational Linguistics: ACL 2025. 2025.

**Questions:**

1. How does the proposed method compare to existing RAG-based KGQA methods?
2. How to alleviate the hallucination issues brought by the LLM rewriter.

---

> ### Author Response · Authors · 2025-11-28
> **Response (1/3)**
>
> Thank you for your time and the insightful comments! Below we address your concerns point by point.
>
> > Limited Novelty: While the retrieval and reasoning framework is novel for KGC, similar frameworks have been widely explored in other similar tasks
>
> The core novelty of PARR lies in using pre-trained interpretable link predictors to provide the missing supervision needed for structural learning in KGC. This supervision enables a fully generative LLM-based model to surpass SoTA discriminative link predictors **for the first time** (to our best knowledge). Our experiments show that the lack of structural supervision is exactly why existing LLM-based link predictors(e.g. [1]) underperform significantly compared to KGE/GNN models. The rewrite–retrieve–reason pipeline is an implementation detail rather than the conceptual contribution (for instance, one would not dismiss a method’s novelty simply because it uses gradient descent).
>
> > ...clarify the differences and connections between their work and prior works in KGQA
>
> While KGC and KGQA share some conceptual similarities, there are significant differences in between, which make them rarely considered under the same domain in existing literature. Below we list the most notable differences:
>
> 1. **Success in KGC depends on modeling relational and structural patterns, not surface-level semantics.** Works like Ultra [2] have shown that encoding pure interactions (without learning any fixed semantics for each entities and relations) can achieve SoTA with generealizable capacity. On the contrary, KGQA is much more semantic-driven, which is why recent KGQA methods lean toward LLM-RAG approaches, while KGC methods remain predominantly KGE/GNN-based.
> 2. **KGC operates on incomplete KGs.** The goal of KGC is to predict missing links, as many relevant paths simply do not exist in the observed KG. This means we cannot obtain ground-truth reasoning paths by simply traversing the KG for typical KGC datasets like WN18RR. On the other hand, KGQA methods, such as RoG and GNN-RAG mentioned by the reviewer, typically directly utilize the ground-truth shortest path obtained by graph traversal.
> 3. **KGC has much less inherent supervision signal.** KGQA datasets typically provides ground-truth retrieval. Moreover, because of 2. mentioned above, it is also easier for KGQA methods to obtain ground-truth reasoning path due to the completeness of the KG. These supervisions are not present in KGC, and are exactly what we believe hinder the native LLM-based KGC methods, and what PARR solves (use pre-trained link predictors to provide **a.** retrieval signal, and **b.** CoT signal for structural learning).
>
> > Comparison with RoG and GNN-RAG
>
> We are not aware of either RoG or GNN-RAG a priori (in the KGC literature, it is less common to compare with KGQA because of the distinct task types discussed above). Below we compare PARR with RoG and GNN-RAG for clarity:
>
> 1. RoG: RoG follows a rewrite–retrieve–reason pipeline and uses extracted reasoning paths to supervise both its planning and reasoning modules. RoG has access to ground-truth retrieval and ground-truth reasoning paths. However, such supervision is unavailable in KGC.
> 2. GNN-RAG: GNN-RAG employs a GNN as a dense subgraph retriever and again relies on gold reasoning paths derived from a complete KG. Like RoG, it assumes the availability of supervision signals that are again not present in KGC.
>
> In summary, even though RoG and GNN-RAG share a high-level retrieve–reason intuition, they operate under different KG types and assumptions (different structuralness and completeness of KG; accessiblity of ground-truth retrieval and reasoning paths) from the KGC literature. PARR’s contribution is precisely in addressing this gap: using interpretable link predictors to provide the missing retrieval and reasoning supervision, enabling a generative LLM to perform competitively on KGC tasks.
>
> We have added a related work section discussing the above aspects.

---

> ### Author Response · Authors · 2025-11-28
> **Response (2/3)**
>
> > Unsatisfied Claims on Hallucination Reduction
>
> We believe there are quite a few misunderstandings, for which we clarify below.
>
> ***"The authors claim that PARR advances previous LLM-based methods by reducing hallucination in prediction through the use of retrieved proofs."***
>
> We kindly clarify this is **not** the case. The "hallucination" we refer to in PARR is: "hallucination in KG reasoning when exposed to noisy or irrelevant retrievals" (L72 in the revised manuscript). The effect of such hallucination reduction is already shown in Fig. 3 and Tab. 5.
>
> ***"However, the proposed rewriter is still purely based on LLMs' internal knowledge learned from training."***
>
> We would like to kindly mention the hallucination we referred to is about the Reasoner's lack of robustness when exposed with noisy retrievals, not about the rewriter.
>
> > Therefore, it still has the potential to generate hallucinated facts that do not exist in the knowledge graph. The authors should provide more empirical evidence to evaluate the rewrite results and support their claim on hallucination reduction.
>
> First, we are **not** claiming we are able to **fully eliminate** hallucination. Second, hallucination reduction is not the main focus of this work. In fact we only mention the term "hallucination" **once** in the entire paper (L72). Third, we have provided extensive evaluation about the reasoner before and after finetuning, demonstrating the effectiveness of PARR. We kindly refer the reviewer to Table 5, Figure 3, Section 5.3 (Ablations on Reasoner), and Appendix C.1 for more details.
>
> To avoid further confusion, we will replace the term "Hallucination" with "Lack of robustness to noisy retrieval" in the final manuscript. We will keep it unchanged for now for your reference.
>
> > Training Complexity: The training cost of PARR is relatively high...making it less practical for large-scale KGs.
>
> 1. Compared to recent methods like KICGPT and MKGL, PARR is indeed more efficient and scalable in training and inference. Unlike KGE or GNN-based methods, where complexity increase at least linear to KG size, PARR has nearly constant inference complexity (below in 3. we show that we can decouple pretrained link predictors away from inference), since retrieval has a fixed budget, and the KG size does not necessarily impact reasoning length. For instance, during training for YAGO3-10, PARR takes 923 hours for both rewriter and reasoner, whereas MKGL takes 3789 hours. During inference for FB15k-237, KICGPT takes 176 seconds per batch, whereas PARR takees 3.5 seconds. Please refer to Table 7 and Table 8 in Appendix A for more details.
> 2. As we mentioned at the very beginning of our introduction (2nd paragraph), we believe the bottleneck for KGC is **not** efficiency: knowledge-intensive domains like healthcare and scientific discovery require deliberate reasoning and interpretability over speed, which PARR excels at. Further more, we believe PARR has many potential future directions. For instance, one of our ongoing effort is to unify different KG tasks (e.g. node classification and CQA) all under one model using PARR in a conversational manner. Therefore, we believe the additional computational is worth it.
> 3. While there seems to be multiple components inside PARR, note that the majority efforts for PARR lie in the data preparation stage (i.e. proof extraction). Here we also provide the GPU hours spent for extracting proofs in the below table. We can see it's actually much more efficient than the widely-used LLM-based synthetic data creation methods. For instance, for WN18RR, with batch size=1, it requires ~4 hours with peack GPU memory < 1G. By parallelizing the process, we can finish in less than an hour.
>
>     | | WN18RR | FB15k237 | YAGO3-10 |
>     |--|--|--|-|
>     | Total GPU Hour  | 0.36   | 1.64     | 6.63     |
>
> 4. To further enhance test-time efficiency, we additionally tested the setting where no proofs are used during retrieval (PARR-Qwen3 w/o proofs), which allows us to decouple pre-trained link predictors entirely from inference stage of PARR. We find this setting leads to little performance degradation, as shown in the below table.
>
>     | Method | FB15k-237 | WN18RR |
>     |-|-|--|
>     | ComplEx+RP| 0.298     | 0.443  |
>     | NBFNet| 0.321     | 0.497  |
>     | MKGL| 0.325     | 0.500  |
>     | PARR-Qwen3| 0.352     | 0.513  |
>     | **PARR-Qwen3 (w/o proofs)**   | 0.346     | 0.508  |
> 5. Additionally, we find that PARR reasoner converges rather quickly, and especially for larger KGs (e.g. YAGO3-10), due to the overlaps in the constructed rewriting and reasoning sets. This means we can flexibly trade training time with minimal performance degradation. We show the result in Figure 4 (Appendix) of the revised manuscript. For instance, for YAGO3-10, we can obtain competitive performance with ~50% of the training.

---

> ### Author Response · Authors · 2025-11-28
> **Response (3/3)**
>
> > Limited Performance and Generalizability: While the proposed framework involves heavy LLMs and training, its performance and generalizability are still limited compared to traditional KGC methods like NBFNet
>
> 1. PARR is a **generative** LLM-based link predictors, trained with language loss. This is fundamentally different from strong baselines (e.g., MKGL) trained with traditional link prediction loss (i.e. BCE), which is **discriminative**. Due to the generative nature, the task is essentially harder for PARR, as it has to learn the distribution over the vast language space.
> 2. The direct baseline of PARR is KG-Llama ([1]), which train LLMs with regular Supervised Finetuning for link prediction, similar to PARR. Compared to KG-Llama, PARR indeed improves significantly by **over 25%**.
> 3. PARR achieves best HITS@1 for **ALL** tested dataset, for both inductive and transductive setting. Given PARR's completely different framework and future potentials, we do not feel the *slight* improvement shall be a major concern.
> 4. We would like to note that what the reviewer views as "small improvements" are not "small" for link prediction. For instance, MKGL (NeurIPS 2024 Spotlight) [2] has less than 0.004 improvement on HITS@1 compared to NBFNet, and has worse performance on HITS@3 and HITS@10 on both datasets it tested. Numerous other works also have much smaller (or even worse) improvements than their predecessors. For instance, on FB15k-237 (compared to NBFNet):
>
>     | Method| HITS@1 | Difference |
>     |--|--|-|
>     | NBFNet | 0.321  | –  |
>     | KICGPT (ACL 2023) | 0.327  | +0.006     |
>     | MKGL (NeurIPS 2024 Spotlight)| 0.325  | +0.004     |
>     | KG-FIT (NeurIPS 2024) | 0.277  | -0.045     |
>     | **PARR (ours)**  | **0.348** | **+0.027** |
>
>     PARR indeed has the largest improvement compared to recent methods.
> 5. We believe PARR has many potentials. For instance, one of our ongoing effort is to unify different KG and KG-related tasks, such as node classification and CQA, into one model under a conversational manner, through the proposed PARR, to build a foundational KG model.
>
> > Moreover, the experiments in inductive settings show that PARR's performance drops significantly, indicating limited generalizability to unseen KGs.
>
> Our experiments in Table 3 and Table 11 clearly shows that **PARR achieves best HITS@1 for ALL tested inductive datasets and splits**. Could the reviewer clarify what it means by *"in inductive settings PARR's performance drops significantly"*?
>
> > Considering the proposed method is built upon the traditional KGC methods like NBFNet
>
> We believe there may be a misconception: just because PARR utilizes traditional KGC methods for data curation does not imply PARR is "built upon" these models. For instance, a large amount of works utilize commercial LLMs (e.g. ChatGPT) for creating synthetic data, but that does not imply these works are all "built on" these LLMs. PARR is different from traditional KGC methods from every perspective: model architecture, training regime, and inference procedure.
>
> > The multi-answer generation setting is not reasonable. Considering the original KG contains 1-to-many relations, it is not necessary to sample top-k predictions from link predictors as ground truth answers.
>
> Thank you for the insightful comment. We indeed consider the 1-to-many relations, and only add additional answers predicted by the pretrained link predictors to head-relation pairs having less than 10 tails. We have revised the manuscript to include more detail in Section 4.5.
>
> > Typos and Notations: The writing and structure of the paper can be improved. There are many short sections that can be merged for better readability and flow. There are also many notation inconsistencies, e.g., using both R to denote relations and retrieved triples.
>
> Thank you for the advice. We have rearranged and merged several short sections for better readability. We have also renamed the retrieval set from $\mathcal{R}$ to $\mathcal{G}$.
>
> [1] Exploring Large Language Models for Knowledge Graph Completion
> [2] Towards Foundation Models for Knowledge Graph Reasoning

---

### Official Review · Reviewer_hZBF · 2025-10-30

**Soundness:** 2
**Presentation:** 1
**Contribution:** 2
**Rating:** 4
**Confidence:** 4

**Summary:**

This paper proposes a large language model (LLM)-based knowledge graph completion (KGC) method. The core idea is to use existing KGC models as retrievers to identify reasoning paths in the graph, and then combine these paths to train an LLM as a reasoner. A mixture-of-experts mechanism is employed to integrate multiple retrievers effectively.
To further enhance the model’s performance, several optimization techniques are incorporated, such as query rewriting, fine-tuning, and sampling strategies. Experimental results demonstrate the effectiveness and superiority of the proposed approach.

**Strengths:**

In the experiments, the authors compare the proposed method with a wide range of baselines, demonstrating that it achieves the best overall performance.

The proposed pipeline is comprehensive, encompassing key components such as query rewriting, retrieval, reasoning, and fine-tuning.

The paper is easy to understand and provides sufficient details to facilitate understanding.

**Weaknesses:**

Although the paper is relatively easy to follow, the writing quality is quite rough and would benefit from substantial revision for clarity and readability.

The experiments are conducted on only three knowledge graphs for the transductive setting and two for the inductive setting. However, prior works typically evaluate on a wider range of datasets, such as Family and UMLS.

In the inductive setting, it is unclear why several common baselines are omitted.

Moreover, the ablation study is not sufficiently comprehensive—there are no experiments analyzing time complexity. The proposed method also appears to be computationally expensive.

**Questions:**

please refer to Weaknesses

---

> ### Author Response · Authors · 2025-11-28
> **Response (1/2)**
>
> Thank you for your time and the insightful comments! Below we address your concerns point by point.
>
> > Although the paper is relatively easy to follow, the writing quality is quite rough and would benefit from substantial revision for clarity and readability.
>
> Thank you for the comment. We have refactored and rearranged multiple sections for better readability. If there are any sections in the manuscript for which you still feel are unclear, please let us know and we are more than happy to revise them.
>
> > The experiments are conducted on only three knowledge graphs for the transductive setting and two for the inductive setting. However, prior works typically evaluate on a wider range of datasets, such as Family and UMLS.
>
> 1. We have tested PARR on four popular, mid-size KGC datasets: FB15k-237, WN18RR, NELL995 and YAGO3-10, as well as the inductive setting on FB15k237 and WN18RR. Compared to numerous prior works, we believe our evaluation is indeed extensive. On the contrary, statistical KGs like UMLS/Nations/Kinship etc. are less commonly used in recent KGC methods due to their limited scales. Below we list the datasets tested by prior works and PARR.
>
>     | Method   | FB15k237 | WN18RR | YAGO3-10 | NELL995 | FB15k237-ind | WN18RR-ind |
>     |----------|----------|--------|----------|---------|---------------|-------------|
>     | RotatE   | ✓        | ✓      | ✗        | ✗       | ✗             | ✗           |
>     | NBFNet   | ✓        | ✓      | ✗        | ✗       | ✓             | ✓           |
>     | KICGPT   | ✓        | ✓      | ✗        | ✗       | ✗             | ✗           |
>     | MKGL     | ✓        | ✓      | ✗        | ✗       | ✓             | ✓           |
>     | KG-Llama | ✗        | ✓      | ✓        | ✗       | ✗             | ✗           |
>     | **PARR**     | ✓        | ✓      | ✓        | ✓       | ✓             | ✓           |
> 2. As mentioned by the reviewer, we further test on three small size KGs: Kinship, Nations and UMLS. We show the table below, where PARR achieves SoTA performance in all cases.
>
>     | Method       | Kinship        |               | UMLS           |               | Nations        |               |
>     |--------------|----------------|---------------|----------------|---------------|----------------|---------------|
>     |              | H@1            | H@10          | H@1            | H@10          | H@1            | H@10          |
>     | NeuralLP     | 0.475          | 0.912         | 0.643          | 0.862         | –              | –             |
>     | MINERVA      | 0.605          | 0.924         | 0.728          | 0.968         | –              | –             |
>     | DRUM         | 0.367          | 0.885         | 0.546          | 0.935         | –              | –             |
>     | NBFNet       | 0.632          | 0.966         | 0.721          | 0.971         | 0.633          | 0.951         |
>     | LERP         | 0.500          | 0.931         | 0.646          | 0.942         | –              | –             |
>     | PARR-Qwen3   | **0.656**          | **0.973**         | **0.748**          | **0.983**         | **0.672**          | **0.960**         |
>
>
>
> 3. We additionally tested PARR on ogbl-wikikg2, a large KG etracted from Wikidata. We show the results (MRR) below. PARR again achieves best performance, despite being the lower bound for MRR (because in PARR we only predict top-10 entities, where entities not appearing in the top-10 will not contribute to MRR calculation).
>
>
>     |                |**PARR-Qwen3** | TransE | ComplEx | RotatE | PairRE | ComplEx+RP | AStarNet  |
>     |----------------|--------|---------|--------|--------|------------|----------|------------|
>     | MRR           | **0.7013** | 0.4256 | 0.4027  | 0.4332 | 0.5208 | 0.6392     | 0.6767        |
>
>
> > In the inductive setting, it is unclear why several common baselines are omitted.
>
> For the inductive setting result in Table 3, we simply follow the reported numbers from MKGL [1]. The fact that some baselines reported in Table 1 are not shown in Table 3 should mostly because these baselines do not report the inductive setting.

---

> ### Author Response · Authors · 2025-11-28
> **Response (2/2)**
>
> > Moreover, the ablation study is not sufficiently comprehensive—there are no experiments analyzing time complexity.
>
> 1. We have indeed provided detailed statistics on time complexity in Appendix A (L648-709).
> 2. In terms of ablations, we believe we have sufficiently analyzed each module, design choice, and robustness of our proposed method. If there are any specific ablations the reviewer feels are necessary, please let us know.
>
> > The proposed method also appears to be computationally expensive.
>
> 1. Compared to recent methods like KICGPT and MKGL, PARR is indeed more efficient in training and inference. Unlike KGE or GNN-based methods, where complexity increase at least linear to KG size, PARR has nearly constant inference complexity (below in 3. we show that we can decouple pretrained link predictors away from inference), since retrieval has a fixed budget, and the KG size does not necessarily impact reasoning length. For instance, during training for FB15k-237, PARR takes 923 hours, where as MKGL takes 3789 hours. During inference for FB15k-237, KICGPT takes 176 seconds per batch, whereas PARR takees 3.5 seconds. Please refer to Table 7 and Table 8 in Appendix A for more details.
> 2. As we mentioned at the very beginning of our introduction (2nd paragraph), we believe the bottleneck for KGC is **not** latency: knowledge-intensive domains like healthcare and scientific discovery require deliberate reasoning and interpretability over inference speed, which PARR excels at. Further more, we believe PARR has many potential future directions. For instance, one of our ongoing effort is to unify different KG tasks (e.g. node classification and CQA) all under one model using PARR. Therefore, we believe the additional computational is worth it.
> 3. To further enhance test-time efficiency, we additionally tested the setting where no proofs are used during retrieval (PARR-Qwen3 w/o proofs), which allows us to decouple pre-trained link predictors entirely from inference stage of PARR. We find this setting leads to little performance degradation, as shown in the below table.
>
>     | Method                     | FB15k-237 | WN18RR |
>     |---------------------------|-----------|--------|
>     | ComplEx+RP                | 0.298     | 0.443  |
>     | NBFNet                    | 0.321     | 0.497  |
>     | MKGL                      | 0.325     | 0.500  |
>     | PARR-Qwen3                | 0.352     | 0.513  |
>     | **PARR-Qwen3 (w/o proofs)**   | 0.346     | 0.508  |
> 4. Additionally, we find that PARR reasoner converges rather quickly, and especially for larger KGs (e.g. YAGO3-10), due to the overlaps in the constructed rewriting and reasoning sets. This means we can flexibly trade training time with minimal performance degradation. We show the result in below table, as well in Figure 4 (Appendix) of the revised manuscript. For instance, for YAGO3-10, we can obtain competitive performance with ~50% of the training.
>
>     |     | 20%    | 40%    | 60%    | 80%    | 100%   |
>     |------------|-------|-------|-------|-------|-------|
>     | WN18RR     | 0.203 | 0.364 | 0.466 | 0.495 | 0.501 |
>     | FB15k-237  | 0.177 | 0.292 | 0.347 | 0.351 | 0.364 |
>     | YAGO3-10   | 0.316 | 0.463 | 0.491 | 0.499 | 0.502 |

---

### Official Review · Reviewer_GZPL · 2025-11-01

**Soundness:** 3
**Presentation:** 2
**Contribution:** 2
**Rating:** 4
**Confidence:** 4

**Summary:**

This paper introduces **PARR (Proof-Augmented Retrieval and Reasoning)** — a framework for knowledge graph completion (KGC) that supervises large language models (LLMs) using *proof paths* extracted from interpretable link predictors such as NBFNet and A*Net:contentReference[oaicite:0]{index=0}.
The framework consists of three modules:
1. **Rewriter** – decomposes queries into sub-queries to improve retrieval coverage.
2. **Retriever** – gathers triplets and associated proofs to form subgraph contexts.
3. **Reasoner** – performs chain-of-thought reasoning to predict missing links.

PARR fine-tunes Llama3 and Qwen3 models on these tasks and reports competitive performance on FB15K-237, WN18RR, and YAGO3-10 datasets in both transductive and inductive settings.

**Strengths:**

- **Bridging neurosymbolic and LLM reasoning.** The integration of interpretable link predictors (proof-based reasoning) with LLMs is conceptually appealing and timely, aligning with the trend toward more structured and interpretable language models.
- **Methodological completeness.** The paper covers the entire pipeline—proof extraction, query rewriting, reasoning fine-tuning, and evaluation—with comprehensive ablation studies that clarify component contributions:contentReference[oaicite:1]{index=1}.
- **Empirical validation.** Results show consistent, if modest, improvements on HITS@1 and HITS@3 metrics over prior LLM-based baselines like MKGL and KICGPT.
- **Readable and organized.** The writing is clear, figures are helpful, and the method is easy to follow even for readers not deeply familiar with KGC literature.

**Weaknesses:**

- **Limited conceptual novelty.** The main ideas—query rewriting, proof-based retrieval, and CoT reasoning—are all well-established in related literature (e.g., RAG, symbolic reasoning, and interpretable KGE). The contribution lies more in *combining* these ideas than introducing a fundamentally new concept.
- **Marginal empirical gains.** Improvements over strong baselines (e.g., MKGL) are small, typically within 1–2% HITS@1:contentReference[oaicite:2]{index=2}. This raises questions about how much the proposed supervision actually improves structural reasoning versus serving as an additional form of data augmentation.
- **Interpretability not demonstrated.** Despite emphasizing “proof-guided interpretability,” the paper provides no qualitative analysis or human-interpretable reasoning traces. Without such evidence, the interpretability claim remains speculative.
- **Dependence on external models.** The reliance on pre-trained interpretable link predictors for proof extraction makes the framework less self-contained and raises scalability concerns for large KGs like Wikidata.
- **Narrow evaluation scope.** Experiments are limited to small, standard KGC datasets. There is no demonstration that PARR generalizes to large-scale or real-world knowledge graphs in scientific or biomedical domains.

**Questions:**

1. How exactly does proof-guided CoT supervision differ from standard reasoning fine-tuning using retrieved triplets?
2. What is the runtime and resource cost of proof extraction and mixture-of-experts (MoE) sampling?
3. Can the framework generalize beyond link prediction—for instance, to multi-hop question answering?
4. Are there cases where noisy or contradictory proofs degrade reasoning performance?

---

> ### Author Response · Authors · 2025-11-28
> **Response (1/3)**
>
> Thank you for your time and the insightful comments! Below we address your concerns point by point.
>
> > Limited conceptual novelty. The main ideas—query rewriting, proof-based retrieval, and CoT reasoning—are all well-established in related literature (e.g., RAG, symbolic reasoning, and interpretable KGE). The contribution lies more in combining these ideas than introducing a fundamentally new concept.
>
> Thank you for the thoughtful comment. We would like to clarify that the core conceptual novelty of PARR does not lie in the individual techniques (query rewriting, proof-based retrieval, or CoT reasoning), which indeed have been explored in prior work. Instead, our main contribution is the new paradigm of using **discriminative**, interpretable link predictors to supervise and guide the generative retrieval and reasoning of LLMs for **generative** link prediction. To the best of our knowledge, PARR is the first work that explores such paradigm and enables generative LLM-based link predictors to match the performance of traditional, discriminative SoTAs.
>
> > Marginal empirical gains. Improvements over strong baselines (e.g., MKGL) are small, typically within 1–2% HITS@1. This raises questions about how much the proposed supervision actually improves structural reasoning versus serving as an additional form of data augmentation.
>
> 1. PARR is a **generative** LLM-based link predictors, trained with language loss. This is fundamentally different from strong baselines (e.g., MKGL) trained with traditional link prediction loss (i.e. BCE), which is **discriminative**. Due to the generative nature, the task is essentially harder for PARR, as it has to learn the distribution over the vast language space.
> 2. The direct baseline of PARR is KG-Llama ([1]), which train LLMs with regular Supervised Finetuning for link prediction, similar to PARR. Compared to KG-Llama, PARR indeed improves significantly by **over 25%**.
> 3. PARR achieves best HITS@1 for **ALL** tested dataset, for both inductive and transductive setting. Given PARR's completely different framework and future potentials, we do not feel the *slight* improvement shall be a major concern.
> 4. Lastly, we would like to note that what the reviewer views as "small improvements" are not "small" for link prediction. For instance, MKGL (NeurIPS 2024 Spotlight) [2] has less than 0.004 improvement on HITS@1 compared to NBFNet, and has worse performance on HITS@3 and HITS@10 on both datasets it tested. Numerous other works also have much smaller (or even worse) improvements than their predecessors. For instance, on FB15k-237 (compared to NBFNet):
>
>     | Method                               | HITS@1 | Difference |
>     |--------------------------------------|--------|------------|
>     | NBFNet                               | 0.321  | –          |
>     | KICGPT (ACL 2023)                    | 0.327  | +0.006     |
>     | MKGL (NeurIPS 2024 Spotlight)        | 0.325  | +0.004     |
>     | KG-FIT (NeurIPS 2024)                | 0.277  | -0.045     |
>     | **PARR (ours)**                      | **0.348** | **+0.027** |
>
> PARR indeed has the largest improvement compared to recent methods.
>
> > Interpretability not demonstrated. Despite emphasizing “proof-guided interpretability,” the paper provides no qualitative analysis or human-interpretable reasoning traces. Without such evidence, the interpretability claim remains speculative.
>
> We have indeed provided qualitative examples showing both the rewrited query and the reasoning process of the Reasoner. Please refer to Figure 5. in Appendix F for the samples. Here we show a sample generation from the reasoner, including the proof path:
>
> ```
>
> --> Query:
> ( thread, hypernym, ? )
> --> Reasoner Generation:
>
> ### Reasoning:
> Based on the retrieved context, we have:
> ('thread', '(inverse)  derivationally related form', 'threader') -> ('thread', '(inverse)
> verb group', 'run') -> ('threader', 'derivationally related form', 'thread')
> Therefore, we know:
> ('thread', 'hypernym', 'run')
> </think>
> ### Answer:run
> ```

---

> ### Author Response · Authors · 2025-11-28
> **Response (2/3)**
>
> > Dependence on external models. The reliance on pre-trained interpretable link predictors for proof extraction makes the framework less self-contained ...
>
> 1. Trends in current AI are towards more and more complex systems.  PARR provides an effectively way to reuse pre-trained simple link predictors to supervise LLMs, thereby integrating thse simple link predictors into the LLM framework.
> 2. We additionally conducted ablations where we train our rewriter based on fixed retriever without retrieving proof (PARR-Qwen3 w/o proofs), which allows us to decouple pre-trained link predictors from inference time, while only use it to generate training data. We find this setting's performance to be on par with our original setting, where proofs are included as part of the KB. We show the results below:
>
>     | Method                     | FB15k-237 | WN18RR |
>     |---------------------------|-----------|--------|
>     | ComplEx+RP                | 0.298     | 0.443  |
>     | NBFNet                    | 0.321     | 0.497  |
>     | MKGL                      | 0.325     | 0.500  |
>     | PARR-Qwen3                | 0.352     | 0.513  |
>     | **PARR-Qwen3 (w/o proofs)**   | 0.346     | 0.508  |
>
>
>
> > ...and raises scalability concerns for large KGs like Wikidata.
>
> 4. We additionally tested PARR on ogbl-wikikg2, a large KG etracted from Wikidata. We show the results (MRR) below. PARR again achieves best performance, despite being the lower bound for MRR (because in PARR we only predict top-10 entities, where entities not appearing in the top-10 will not contribute to MRR calculation).
>     | Metric | **PARR-Qwen3** | TransE | ComplEx | RotatE | PairRE | ComplEx+RP | AStarNet |
>     |--------|-----------------|--------|---------|--------|---------|------------|----------|
>     | MRR    | **0.7013**      | 0.4256 | 0.4027  | 0.4332 | 0.5208  | 0.6392     | 0.6767   |
>
>
>
> > Narrow evaluation scope. Experiments are limited to small, standard KGC datasets. There is no demonstration that PARR generalizes to large-scale or real-world knowledge graphs in scientific or biomedical domains.
>
> 1. We have tested PARR on four popular, mid-size KGC datasets: FB15k-237, WN18RR, NELL995 and YAGO3-10, as well as the inductive setting on FB15k237 and WN18RR. Compared to numerous prior works, we believe our evaluation is indeed extensive. Below we list the datasets tested by prior works and PARR.
>
>     | Method   | FB15k237 | WN18RR | YAGO3-10 | NELL995 | FB15k237-ind | WN18RR-ind |
>     |----------|----------|--------|----------|---------|---------------|-------------|
>     | RotatE   | ✓        | ✓      | ✗        | ✗       | ✗             | ✗           |
>     | NBFNet   | ✓        | ✓      | ✗        | ✗       | ✓             | ✓           |
>     | KICGPT   | ✓        | ✓      | ✗        | ✗       | ✗             | ✗           |
>     | MKGL     | ✓        | ✓      | ✗        | ✗       | ✓             | ✓           |
>     | KG-Llama | ✗        | ✓      | ✓        | ✗       | ✗             | ✗           |
>     | PARR     | ✓        | ✓      | ✓        | ✓       | ✓             | ✓           |
>
> 2. We additionally evaluate on ogbl-wikikg2, where we show PARR also achieve SoTA performance. Please see our response to your concern (*"…and raises scalability concerns for large KGs like Wikidata."* ) directly above.
> 3. We further test on three small size KGs: Kinship, Nations and UMLS. We show the table below, where PARR achieves SoTA performance in all cases.
>
>     | Method       | Kinship        |               | UMLS           |               | Nations        |               |
>     |--------------|----------------|---------------|----------------|---------------|----------------|---------------|
>     |              | H@1            | H@10          | H@1            | H@10          | H@1            | H@10          |
>     | NeuralLP     | 0.475          | 0.912         | 0.643          | 0.862         | –              | –             |
>     | MINERVA      | 0.605          | 0.924         | 0.728          | 0.968         | –              | –             |
>     | DRUM         | 0.367          | 0.885         | 0.546          | 0.935         | –              | –             |
>     | NBFNet       | 0.632          | 0.966         | 0.721          | 0.971         | 0.633          | 0.951         |
>     | LERP         | 0.500          | 0.931         | 0.646          | 0.942         | –              | –             |
>     | PARR-Qwen3   | **0.656**          | **0.973**         | **0.748**          | **0.983**         | **0.672**          | **0.960**         |
>
> > How exactly does proof-guided CoT supervision differ from standard reasoning fine-tuning using retrieved triplets?
>
> They are the same thing: we use standard cross-entropy loss in LLM finetuning (L249-258).

---

> ### Author Response · Authors · 2025-11-28
> **Response (3/3)**
>
> > What is the runtime and resource cost of proof extraction and mixture-of-experts (MoE) sampling?
>
> Thank you for the question. Below we show the total GPU hour on Nvidia H100 GPU for one expert:
>
> |                 | WN18RR | FB15k237 | YAGO3-10 |
> |-----------------|--------|----------|----------|
> | Total GPU Hour  | 0.36   | 1.64     | 6.63     |
>
> For WN18RR and FB15k237, we use NBFNet and AStarNet, each with two variants, resulting in 4x runs for proof sampling. For YAGO3-10, we only use AStarNet with 2 variants, resulting in 2x runs for proof sampling.
>
> We can see it's actually much more efficient than the widely-used LLM-based synthetic data creation methods. For instance, for WN18RR, with batch size=1, it requires ~4 hours with peack GPU memory < 1G. By parallelizing the process, we can finish in less than an hour.
>
> > Can the framework generalize beyond link prediction—for instance, to multi-hop question answering?
>
> Thank you for the thoughtful question. PARR can flexibly generalize to **any** KG-related tasks (e.g. node classification, CQA etc.), thanks to the general training and inference framework of PARR. This is indeed our ongoing project (i.e. unifying different KG tasks under one model). However, since CQA are typically separated from simple link prediction in prior works, we feel it might be out-of-scope to include in the current PARR paper.
>
> > Are there cases where noisy or contradictory proofs degrade reasoning performance?
>
> Thank you for the thoughtful question! We answer in two folds:
>
> 1. **Training:** It is possible for the noise/bias from the pre-trained link predictors to propagate to the downstream reasoner. To prevent such case, we:
>     - We filter out low-quality proofs based on their path scores and avoid duplications/cycles.
>     - We use a mixture-of-experts to mitigate initialization/model bias. We ablate this effect in Table 6.
> 2. **Inference:**
>     - In general, we do not observe PARR reasoner failing due to extracted noisy proofs at test time. Most failure we observe is due to incomplete retrievals.
>     - To test for the reasoner's robustness to noisy proof (i.e. incorrect proofs), we conduct an ablation (Section C.1, L760-768) where we add $n$% incorrect proofs to the retrieval. We observe the reasoner exhibits decent robustness to such noise, dropping only 3% when 50% noisy proofs are added.
>
> [1] Exploring Large Language Models for Knowledge Graph Completion
>
> [2] MKGL: Mastery of a Three-Word Language

---

### Author Response · Authors · 2025-11-28

Dear Reviewers,

Thank you for all your time and thoughtful advice. Below we list the changes to the manuscript.

1. Conducted Additional experiments on Kinship/UMLS/Nations, and large-scale KG ogbl-wikidata2, where PARR achieves best HITS@1 for **all** the datasets.
2. Improved readability, fixed typos/math notations.
3. Added missing related work.
4. Added analysis on model convergence.
5. Tested additional setting to decouple pretrained link predictors from inference stage, for improved inference efficiency.

---

### Meta-Review · Area_Chair_qgjS · 2026-01-07

**Summary:**

This paper proposes a large language model (LLM)-based knowledge graph completion (KGC) method. The core idea is to use existing KGC models as retrievers to identify reasoning paths in the graph, and then combine these paths to train an LLM as a reasoner. Four reviewers provided mixing feedbacks on this paper. Reasons leaning for acceptance include novel combination of sumbolic reasoning with generative capability of LLMs, comprehensive evaluation, and good presentation. Concerns include missing baselines, limited minor improvements, novelty of the proposed methods, and efficiency. Some concerns were resolved in the rebuttal phase. Despite of detailed response during the rebuttal phase, some major concerns still remain unresolved. Therefore, I recommend rejection of this paper.

**Reviewer Concerns:**

Reviewer GZPL:

W1: novelty, partially addressed. The authors clarified the novelty but did not provide strong evidence.
W2: minor improvements, the authors provided additional results and the improvements looks satisfying.
W3: interpretability, partially addressed. The authors provided some clarifications but did not provide strong evidence.
W4: Dependence on external models and scalability, seems scalability evaluation and comparison is not provided.
W5: Narrow evaluation scope, addressed, as ogbl-wikikg2 is added.

Reviewer hZBF:

W1: readibility, partially addressed.
W2: evaluation scope, addressed, as some additional datasets are added.
W3: inductive setting, partially addressed, while authors can rerun baselines and report some missing results in the inductive settings, instead of fully relying on previous reported results.
W4: missing ablations, addressed.

Reviewer tbAJ:

W1: novelty, partially addressed. The authors clarified the novelty but did not provide strong evidence.
W2: hallucination reduction, not addressed. The authors seem to define Hallucination Reduction differently from common practice, especially in NLP community. A better definition and evaluation is needed.
W3: training efficiency, not addressed. seems training effiency evaluation and comparison is not provided.
W4: minor improvements, the authors provided additional results and the improvements looks satisfying.

Reviewer cL8A:

W1: Dependence on external models, seems not fully addressed.
W2: source of improvements, partially addressed.

**Reviewer Scores:**

Given that some major concerns still remain unresolved, I think the reviewers' scores remain the same.

---

### Decision · Program_Chairs · 2026-01-26

Reject